## REPORT

# Talin rod domain–containing protein 1 (TLNRD1) is a novel actin-bundling protein which promotes filopodia formation

Alana R. Cowell[1], Guillaume Jacquemet[2,3], Abhimanyu K. Singh[1], Lorena Varela[1], Anna S. Nylund[2,3], York-Christoph Ammon[4], David G. Brown[1], Anna Akhmanova[4], Johanna Ivaska[2,5], and Benjamin T. Goult[1]

**Talin is a mechanosensitive adapter protein that couples integrins to the cytoskeleton. Talin rod domain–containing protein 1 (TLNRD1) shares 22% homology with the talin R7R8 rod domains, and is highly conserved throughout vertebrate evolution, although little is known about its function. Here we show that TLNRD1 is an α-helical protein structurally homologous to talin R7R8. Like talin R7R8, TLNRD1 binds F-actin, but because it forms a novel antiparallel dimer, it also bundles F-actin. In addition, it binds the same LD motif–containing proteins, RIAM and KANK, as talin R7R8. In cells, TLNRD1 localizes to actin bundles as well as to filopodia. Increasing TLNRD1 expression enhances filopodia formation and cell migration on 2D substrates, while TLNRD1 down-regulation has the opposite effect. Together, our results suggest that TLNRD1 has retained the diverse interactions of talin R7R8, but has developed distinct functionality as an actin-bundling protein that promotes filopodia assembly.**

## Introduction

Talin rod domain–containing protein 1 (TLNRD1) is an evolutionarily conserved yet little studied protein that shares homology with the cytoskeletal protein talin. TLNRD1 was originally named mesoderm development candidate 1 (MESDC1) because the gene, located on human chromosome 15, mapped to the *mesd* locus essential for mesoderm development (Holdener et al., 1994; Wines et al., 2001). However, this assignment proved erroneous (Hsieh et al., 2003), and the gene was renamed TLNRD1 (Yates et al., 2017), reflecting its similarity to the R7R8 rod domain of talin.

Talin1 and 2 are cytoplasmic adapters that provide a direct mechanosensitive link between the integrin receptors and the actin cytoskeleton (Calderwood et al., 2013; Goult et al., 2018). Talins are comprised of an N-terminal FERM domain (Elliott et al., 2010) coupled via a flexible linker to a large rod domain comprised of 13 helical bundles, R1–R13 (Fig. 1 A; Goult et al., 2013). 12 of the rod domains are arranged linearly, end to end, to create the large extended talin rod that unfolds in response to mechanical force (Yao et al., 2016). However, the R7R8 rod domains adopt a unique fold, where R8, a four-helix bundle, is inserted into a loop between two helices of the R7 five-helix

bundle (Gingras et al., 2010), creating a branch in the talin rod (Fig. 1, A and B).

Talin R7R8 is emerging as a major signaling hub, linking diverse cytoskeletal elements together (Goult et al., 2018). It forms part of a major actin-binding site (actin-binding site 2 [ABS2]) that spans R4–R8 (Atherton et al., 2015; Kumar et al., 2016; Hemmings et al., 1996). Talin R7R8 also binds to multiple other ligands, many of which contain leucine-aspartate motifs (LD motifs; Alam et al., 2014). R8 binds (i) Rap1-GTP–interacting adaptor molecule (RIAM), implicated in recruitment of talin to the leading edge of cells (Chang et al., 2014; Goult et al., 2013) and to filopodial protrusions (Lagarrigue et al., 2015); (ii) deleted in liver cancer 1 (DLC1; Zacharchenko et al., 2016; Li et al., 2011); (iii) paxillin (Zacharchenko et al., 2016); and (iv) cyclin-dependent kinase 1 (CDK1; Gough et al., 2021). R8 also contains a binding site for vinculin (Gingras et al., 2010) and the intermediate filament protein α-synemin (Sun et al., 2008). The R7 domain binds to KANK proteins (Bouchet et al., 2016; Sun et al., 2016), which serve as platforms for the assembly of large cortical microtubule stabilizing complexes that capture microtubules at the periphery of integrin adhesion complexes (Bouchet et al.,

[1]School of Biosciences, University of Kent, Canterbury, UK; [2]Turku Centre for Biotechnology, University of Turku and Åbo Akademi University, Turku, Finland; [3]Faculty of Science and Engineering, Cell Biology, Åbo Akademi University, Turku, Finland; [4]Cell Biology, Neurobiology and Biophysics, Department of Biology, Faculty of Science, Utrecht University, Utrecht, The Netherlands; [5]Department of Biochemistry, University of Turku, Turku, Finland.

Correspondence to Benjamin T. Goult: B.T.Goult@kent.ac.uk; A.K. Singh's present address is Rega Institute for Medical Research, Katholieke Universiteit Leuven, Leuven, Belgium.

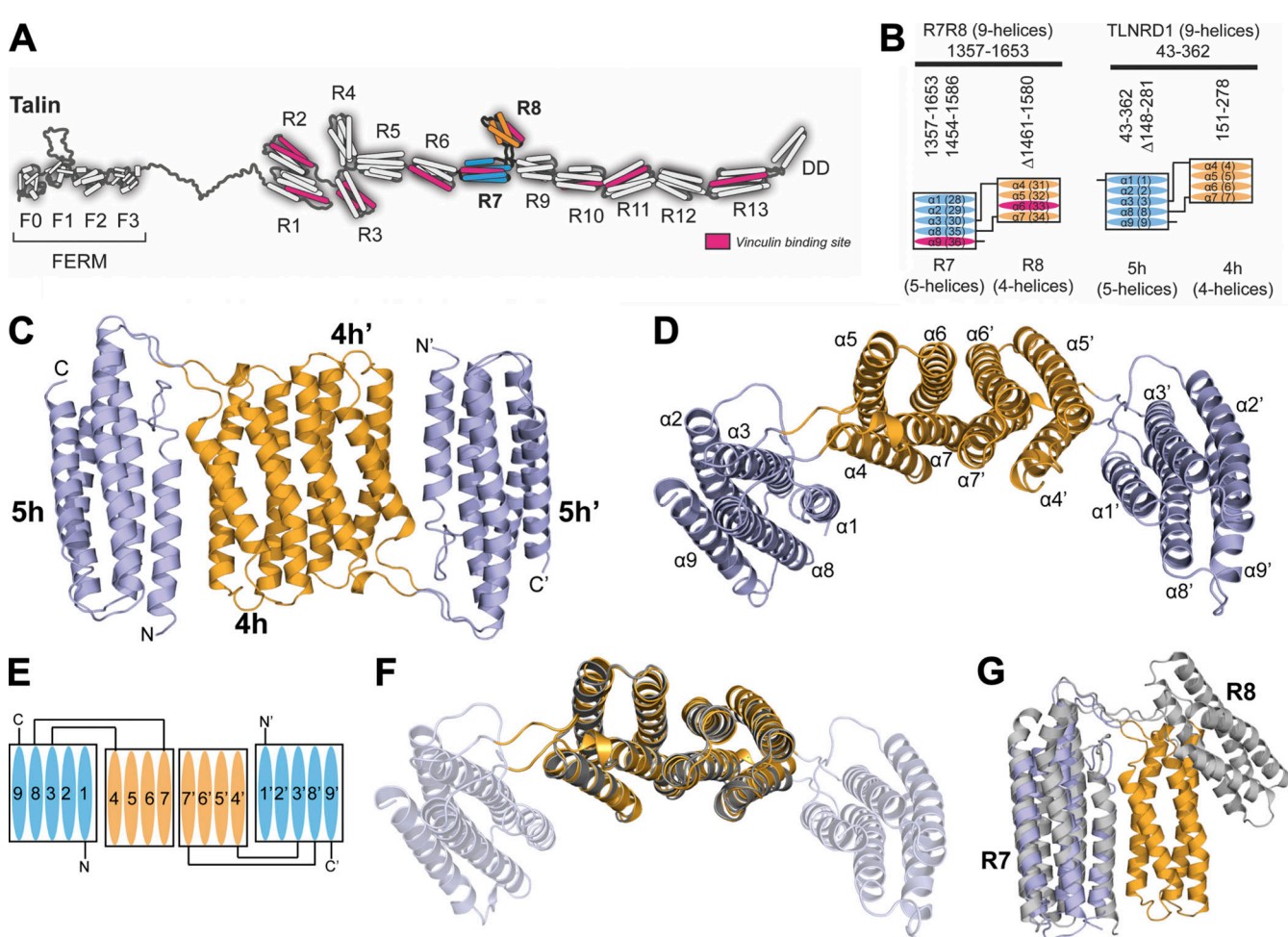

Figure 1. **TLNRD1 is homologous to talin R7R8. (A)** The domain structure of talin. Talin has an N-terminal head domain and a large rod domain comprised of 13 helical bundles, R1–R13 (Goult et al., 2013). The R7 (blue) and R8 (orange) domains are highlighted. The 11 VBSs are shown in pink. **(B)** Schematic of talin R7R8 (left) and TLNRD1 (right); the four-helix bundle (orange) is inserted into the loop between α3 and α8 helices of the five-helix bundle (blue). The VBS consensus sequence (Gingras et al., 2010) in talin R7 and R8 (pink) is absent from TLNRD1. **(C)** Crystal structure of TLNRD1-FL reveals an antiparallel, symmetric dimer. The domains of each monomer are labeled as 4h and 5h, and 4h' and 5h'. **(D)** Top-down view of C with the helices labeled. The two four-helix bundles dimerize via the extensive interface between helices α6 and α7. **(E)** Schematic representation of the TLNRD1 dimer. **(F)** Overlay of the TLNRD1-4H structure (orange) on top of TLNRD1-FL (partially transparent) showing the dimerization interface is identical in both. **(G)** Overlay of one monomer of TLNRD1 (orange and blue) with talin R7R8 (gray; Protein Data Bank accession no. 2X0C). The individual domains are structurally homologous, but the orientation relative to each other is different, with talin R7R8 having a more open structure.

2016). This complex protein interaction network suggests that R7R8 plays a key role in coordinating multiple processes including crosstalk between all three cytoskeletal networks. Mutations in R7R8 have been shown to perturb this coordination and increase invasion and migration in cells (Azizi et al., 2021).

Previous studies have shown that TLNRD1 is directly targeted by anti-oncogenic miRNAs, with TLNRD1 overexpression being associated with increased proliferation and xenograft growth in hepatocellular carcinoma (Tatarano et al., 2012; Wu et al., 2017). In contrast, TLNRD1 depletion reduced bladder cancer cell viability, migration, and invasion, and data from multiple databases (Nagy et al., 2018) show that high TLNRD1 mRNA levels often correlate with poor lung cancer patient survival.

Here we show that although TLNRD1 and talin R7R8 both have the same domain structures and topology, and both bind actin, TLNRD1 is unique in its ability to bundle F-actin. Our

structural data show that this is because it dimerizes via its four-helix module. In U2OS cells, TLNRD1 localizes to thick actin stress fibers and filopodial protrusions, with distinctive localization to the filopodia. TLNRD1 overexpression increases filopodia formation and cellular migration in 2D. Finally, we establish biochemically that TLNRD1 can interact with known R7R8 LD motif–containing ligands, potentially impacting their ability to interact with talin and therefore talin function in cells.

## Results

### The origin of TLNRD1

Sequence homology predicts that TLNRD1 has a structure similar to talin R7R8 with an additional N-terminal unstructured region (residues 1–43; Fig. 1 B and Fig. S1 A). This, plus their 22% amino acid sequence identity, suggests that TLNRD1 may have

originated from a gene duplication event. A striking feature is that TLNRD1 is highly conserved throughout animal evolution, first appearing in sponges and choanoflagellates, but absent from Cnidaria, nematodes, and arthropods (Fig. S1 B). While the R7R8 domains of the two mammalian talin isoforms are each encoded by 11 exons, the TLNRD1 gene consists of a single large exon, suggesting that it may have originated from a section of talin mRNA inserted back into an ancestral genome.

## Crystal structure of TLNRD1

To determine the extent of the structural similarity between TLNRD1 and talin R7R8, we used x-ray crystallography to solve the structures of both full-length TLNRD1 (TLNRD1-FL) and the four-helix domain (TLNRD1-4H). This confirmed that TLNRD1 consists of a nine-helix module comprised of a four-helix and five-helix bundle connected via the same unusual domain linkage identified in talin R7R8, where the four-helix bundle (R8) is inserted into a loop between helices α3 and α4 of the five-helix bundle (R7; Fig. 1, C–F). The first 40 residues of TLNRD1-FL are not visible in the density, supporting the secondary structure prediction that the N-terminal region is unstructured.

## TLNRD1 is a symmetric antiparallel dimer mediated by the four-helix bundle

Our previous gel filtration analysis suggested that TLNRD1 is a dimer (Gingras et al., 2010). The TLNRD1 structures reveal the basis for TLNRD1 dimerization, which is mediated by a novel interface on the four-helix bundle. In both the TLNRD1-FL and TLNRD1-4H structures, the four-helix bundle forms an extensive interaction with a four-helix bundle of a second TLNRD1 molecule (Fig. 1, C–F). An identical dimer interface was observed in both the TLNRD1-FL and the TLNRD1-4H structures (Fig. 1 F), which crystallized in different space groups (Table S1). Analysis of the macromolecular interface using PISA (Krissinel and Henrick, 2007) verified that this was a bona fide dimer.

Dimerization of TLNRD1 is mediated via an extensive hydrophobic interface on helices α6 and α7, with the two F250 side chains docking into the opposing molecule (Fig. 2, A and B). The surface of each four-helix bundle has a pocket created by F270′ and the small side chains of G217′ and A260′ that the F250 aromatic rings docks into (Fig. 2 B). As it is a symmetric dimer, the F250′ docks into the equivalent pocket on the other molecule, leading to an antiparallel configuration. The interface is capped at either end by electrostatic interactions between the side chains of E267 and R246′ and vice versa. Analysis of TLNRD1 conservation using the program ConSurf (Ashkenazy et al., 2016) reveals that these residues are highly conserved (Fig. S1 C).

We used size-exclusion chromatography multi-angle light scattering (SEC-MALS) to explore the oligomeric state of TLNRD1 in solution. Both TLNRD1-FL (labeled "a" in Fig. 2 C) and TLNRD1-4H were dimeric at 25°C, and no monomer peak was present with either construct, suggesting a high-affinity dimer (Fig. 2, C and D). TLNRD1-FL also showed a smaller tetramer peak (labeled "b" in Fig. 2 C), suggesting the presence of a dimer of dimers species. To explore the importance of F250 in dimerization, we generated a point mutant of F250 that swapped the aromatic ring for a charged aspartate, F250D (TLNRD1-F250D).

The TLNRD1-F250D mutant was a stable, folded protein suitable for biochemical analyses as judged by circular dichroism (CD; Fig. S2 A), nuclear magnetic resonance (NMR; Fig. S2 B), and SEC (Fig. 2 C), although the melting temperature, $T_m$, determined using CD, was reduced (TLNRD1-FL $T_m$ 69.7°C, TLNRD1-F250D $T_m$ 48°C; Fig. S2 C). Analysis of the F250D mutant using SEC-MALS showed a clear transition from a dimeric to monomeric state (peak "c"), with only a small proportion remaining as a dimer (peak "d"), confirming the importance of F250 in mediating dimerization.

The absence of a TLNRD1 monomer peak on SEC-MALS suggests that dimerization is mediated by a high-affinity interaction. We therefore used microscale thermophoresis (MST) to study the monomer-dimer equilibrium. MST is an established biochemical assay for studying dimerization (Seidel et al., 2013; Lin et al., 2012). In this experiment, unlabeled TLNRD1-FL was titrated against fluorescently tagged TLNRD1-FL and an apparent equilibrium dimer dissociation constant ($K_{d,dimer}$) of 80 nM obtained (Fig. 2 E). MST confirmed that the F250D mutant prevented dimerization, as no $K_d$ was generated. The high affinity suggests that TLNRD1 exists as an obligate dimer, and the F250D mutant renders TLNRD1 monomeric.

## TLNRD1 retains the functionality of talin R7R8

Given the diverse talin R7R8 interactome, we asked whether TLNRD1 binds the same ligands. We first tested RIAM and KANK1 as both contain well-defined LD motif talin binding sites, with RIAM binding to talin R8, and KANK to R7. Fluorescence polarization studies showed that TLNRD1-FL (Fig. 3 A) interacts with RIAM (residues 4–30) with a $K_d$ of 0.25 µM (Fig. 3 B), more tightly than talin R7R8 (Fig. 3 D). TLNRD1-4H bound to RIAM with similar affinity ($K_d$ 0.59 µM; Fig. 3 C), confirming the RIAM interaction is mediated by the four-helix domain. Furthermore, TLNRD1-F250D also bound RIAM, confirming the integrity of the LD binding surface on the four-helix domain in the monomeric form (Fig. S2 D). Lamellipodin, a paralogue of RIAM, also interacted with TLNRD1-4H, albeit with lower affinity ($K_d$ 9 µM; Fig. 3 C).

The KANK1 "KN domain" LD motif (residues 30–60) bound to TLNRD1-FL ($K_d$ 11.6 µM; Fig. 3 E), albeit with substantially lower affinity than talin R7R8 ($K_d$ 0.35 µM; Fig. 3 G). KANK1 binding to talin R7 requires the LDLD sequence in the KN domain, with a quadruple mutation to alanine (KANK1-4A) abolishing binding. Similarly, this KANK1-4A mutant reduced binding to TLNRD1-FL, confirming the interaction is LD-dependent. Last, no interaction was observed between KANK1 and TLNRD1-4H (Fig. 3 F), confirming the TLNRD1-KANK1 interaction is mediated via the five-helix module. The structural information from equivalent talin complexes was used to model the RIAM- and KANK-binding sites onto the TLNRD1-FL structure (Fig. 3 H). Collectively, this biochemical analysis confirms that TLNRD1 has retained the LD binding sites in both the four- and five-helix modules, and by inference, that talin was able to bind LD motifs before the duplication event.

Another major role of talin R7R8 is to bind actin (Hemmings et al., 1996), and TLNRD1 has previously been shown to bind actin (Gingras et al., 2010). Using a high-speed cosedimentation

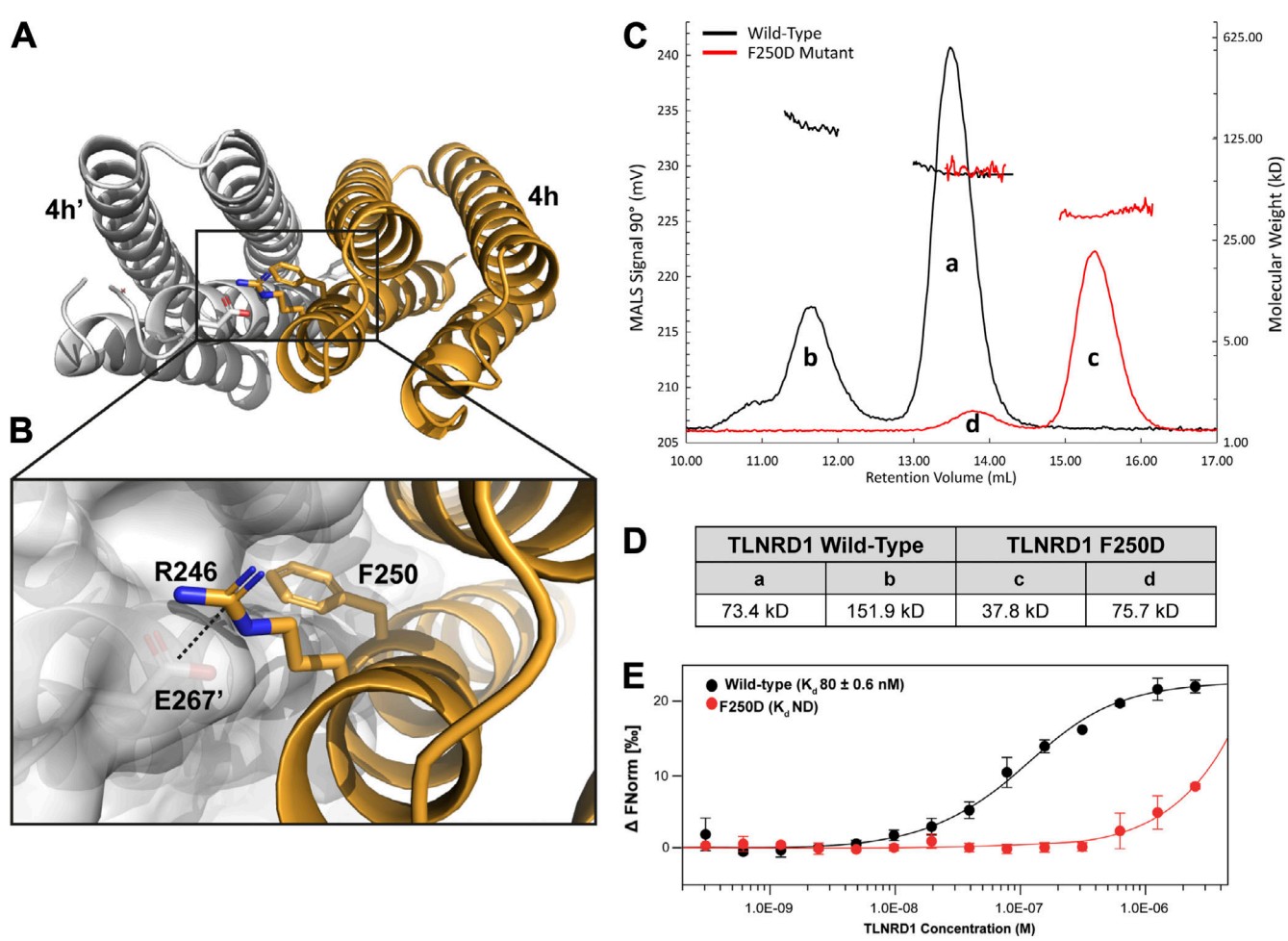

Figure 2. **Characterization of the TLNRD1 dimer. (A and B)** The symmetrical antiparallel TLNRD1 dimer interface mediated by F250. One monomer is shown as a white surface representation, F250 from the opposing monomer (orange) inserts into the pocket on the surface of the other. A salt bridge between R246 and E267' is shown. **(C and D)** SEC-MALS analysis of 106 µM TLNRD1-FL WT (black) and F250D (red). **(D)** Analysis of the molar mass of the major TLNRD1-FL species "a" yields a molecular weight of ~74 kD. The TLNRD1-F250D yields a molecular weight of ~37.8 kD, peak "c." **(E)** TLNRD1-FL dimerization measured using MST at 25°C with 40% Red LED laser excitation. Unlabeled TLNRD1-FL was titrated into a fixed concentration (50 nM) of fluorescently labeled TLNRD1-FL. Data were analyzed in MO.Affinity Analysis software (v2.1.3) using the law of mass action to generate an apparent equilibrium dimer $K_d$, $K_{d,dimer}$ of 80 ± 0.6 nM. F250D $K_{d,dimer}$ not determined. FNorm, normalized fluorescence; ND, not defined.

assay, we confirmed that TLNRD1-FL interacts with actin filaments (Fig. 3 I). Furthermore, we found that TLNRD1-4H alone interacts tightly with actin (Fig. 3 I). This is surprising as, by itself, the equivalent four-helix module in talin, R8, shows little actin binding in cosedimentation assays (Gingras et al., 2010), suggesting that TLNRD1 has enhanced actin filament binding. Taken together, these data indicate that TLNRD1 is an actin-binding protein that has retained functional similarities to talin R7R8.

**TLNRD1 is an actin-bundling protein**

In talin, ABS2 maps to domains R4–R8, with the two four-helix bundles, R4 and R8, engaging the actin filament (Fig. 4 A; Atherton et al., 2015). The actin-binding surface on R8 maps onto one face (helices α2 and α3) of the bundle. Our initial hypothesis was that as TLNRD1 was a dimer, it might engage a single actin filament in a similar fashion to talin ABS2. However, the structure of TLNRD1 revealed that the putative actin-binding surfaces on α2 and α3 are positioned facing away from the dimer interface (Fig. 4 A), raising the possibility that TLNRD1 might engage two actin filaments simultaneously, and thus cross-link and bundle actin.

To establish whether TLNRD1 is also an actin-bundling protein, we used a low-speed (10,000 rpm) cosedimentation assay where actin filaments remain in the supernatant and only bundled actin filaments sediment. Addition of TLNRD1-FL resulted in a clear increase in the levels of actin in the low-speed pellet, confirming that TLNRD1 is an actin bundler (Fig. 4 B). In contrast, neither talin R7R8, nor ABS2, which are both monomeric, is able to bundle actin (Atherton et al., 2015). Given the discovery that TLNRD1-4H can bind actin filaments (Fig. 3 I), we tested the ability of TLNRD1-4H to bundle actin, and surprisingly, we found that the four-helix domain alone is an effective actin bundler (Fig. 4 C and Fig. S2 E). To confirm TLNRD1 bundling activity, actin filaments were visualized by negative stain EM both in the absence (Fig. S2 F) or presence of TLNRD1-FL,

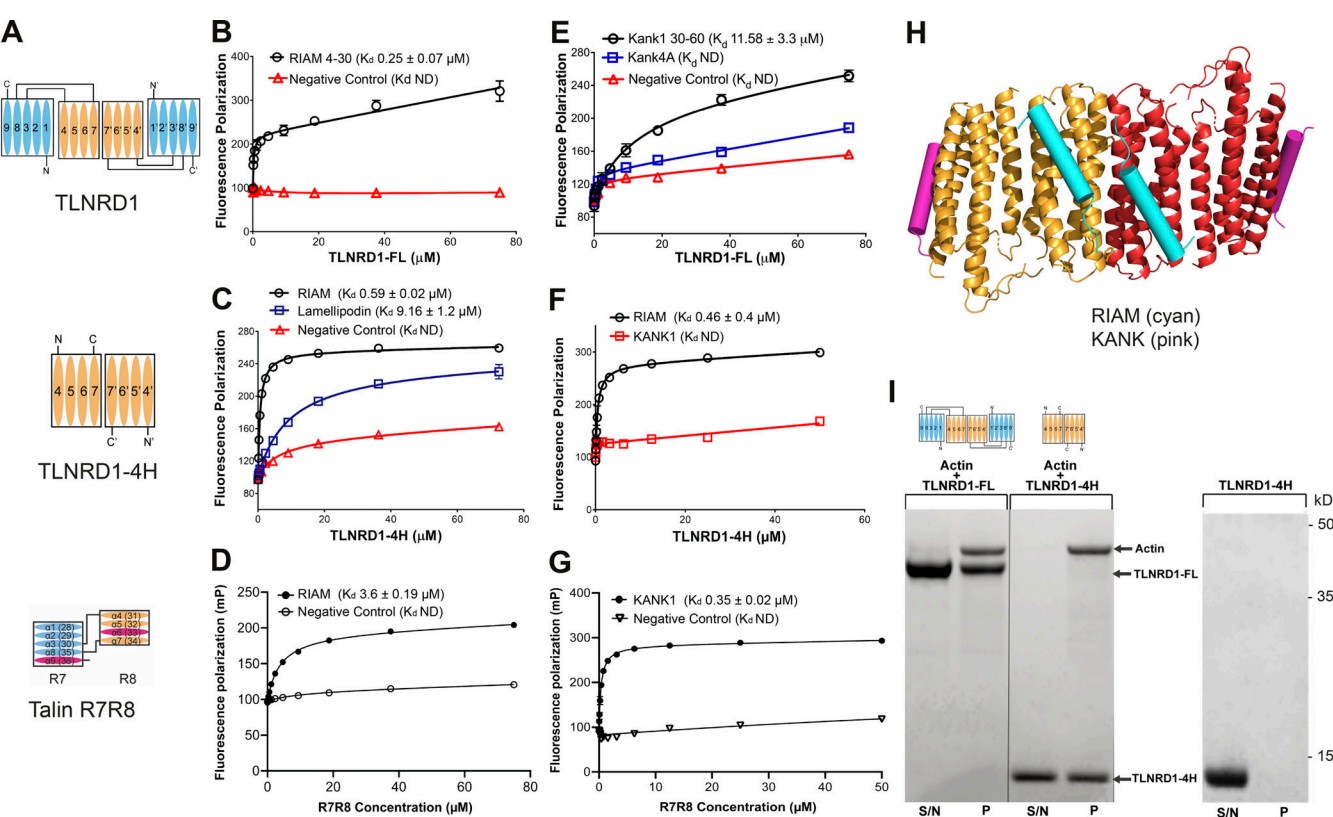

**Figure 3. TLNRD1 and talin R7R8 both bind LD motif proteins and actin. (A)** Schematics of TLNRD1-FL, TLNRD1-4H, and talin R7R8 domain structures. **(B–G)** TLNRD1 binds to LD motif–containing ligands. **(B–D)** Binding of fluorescein-labeled RIAM(4–30) peptide with (B) TLNRD1-FL, (C) TLNRD1-4H, and (D) talin R7R8 measured using a fluorescence polarization assay. The binding of lamellipodin (20–46; blue) is also shown. **(E–G)** BODIPY-labeled KANK1(30–60); wild-type (black) and 4A mutant (blue) peptides (E) binding to TLNRD1-FL, (F) not binding to TLNRD1-4H, and (G) binding to talin R7R8. ND, not defined. Experiments performed in triplicate. **(H)** Structural model of TLNRD1-FL bound to RIAM (cyan) and KANK (pink) LD motifs. **(I)** High-speed actin cosedimentation assay showing that both TLNRD1-FL and TLNRD1-4H interact with F-actin. TLNRD1-4H alone does not pellet. S/N, supernatant; P, pellet.

TLNRD1-4H, and TLNRD1-F250D (Fig. 4, D–F). This revealed that TLNRD1-FL can form large bundles of actin filaments with tight inter-filament spacing and that the 4H domain alone is sufficient for bundling. Similarly, TLNRD1 was found to decorate actin stress fibers when expressed as N-terminally GFP-tagged (GFP-TLNRD1) in U2OS cells (Fig. 4 and Video 1).

A key feature that distinguishes TLNRD1 from talin R7R8 is its dimeric state. To explore the importance of TLNRD1 dimerization in promoting bundling of actin filaments, actin cosedimentation assays were performed with the monomeric TLNRD1-F250D. TLNRD1-F250D was still able to bind to actin filaments (Fig. S2 G), but it showed significantly reduced actin-bundling activity (Fig. S2 H), confirming that dimerization is required for TLNRD1 to bundle actin effectively. This loss of bundling was visualized by EM, where the monomeric TLNRD1-F250D decorates actin filaments (Fig. 4 F) but can no longer bundle them. In summary, TLNRD1 binds actin considerably more tightly than talin R7R8 or ABS2, and also bundles actin.

### TLNRD1 promotes filopodia formation and modulates cell migration

As actin-bundling proteins contribute to filopodia functions (Gupton and Gertler, 2007; Khurana and George, 2011; Jacquemet et al., 2015), we wondered whether TLNRD1 might

also modulate filopodia formation. Overexpression of GFP-TLNRD1 in U2OS promoted MYO10-positive filopodia formation, whereas the expression of GFP-TLNRD1-F250D did not, indicating that TLNRD1 dimerization is required to promote filopodia formation (Fig. 5, A and B). Conversely, silencing TLNRD1 expression using two independent siRNAs (Fig. 5, C and D) led to a decrease in the number of MYO10 filopodia in U2OS cells (Fig. 5 E).

To validate that TLNRD1 also modulates filopodia in the absence of MYO10 overexpression, two models of endogenous filopodia formation were tested. These include (i) U2OS cells actively spreading on fibronectin, and (ii) RAT-2 cells, which readily generate numerous endogenous filopodia (Jacquemet et al., 2019). In both cases, overexpression of TLNRD1 led to a significant increase in filopodia number, while the monomeric mutant TLNRD1-F250D failed to influence filopodia in either cell type (Fig. 5, F–I), indicating that the ability of TLNRD1 to induce filopodia is not secondary to MYO10 overexpression. Imaging GFP-TLNRD1 together with MYO10 showed that TLNRD1 localized to filopodia (Fig. 5 A). To gain further insights into the spatial distribution of TLNRD1 and MYO10 filopodia, cells were imaged using structured-illumination microscopy (Fig. S3, A and B). Surprisingly, these experiments revealed that TLNRD1, unlike other filopodia bundling proteins such as fascin,

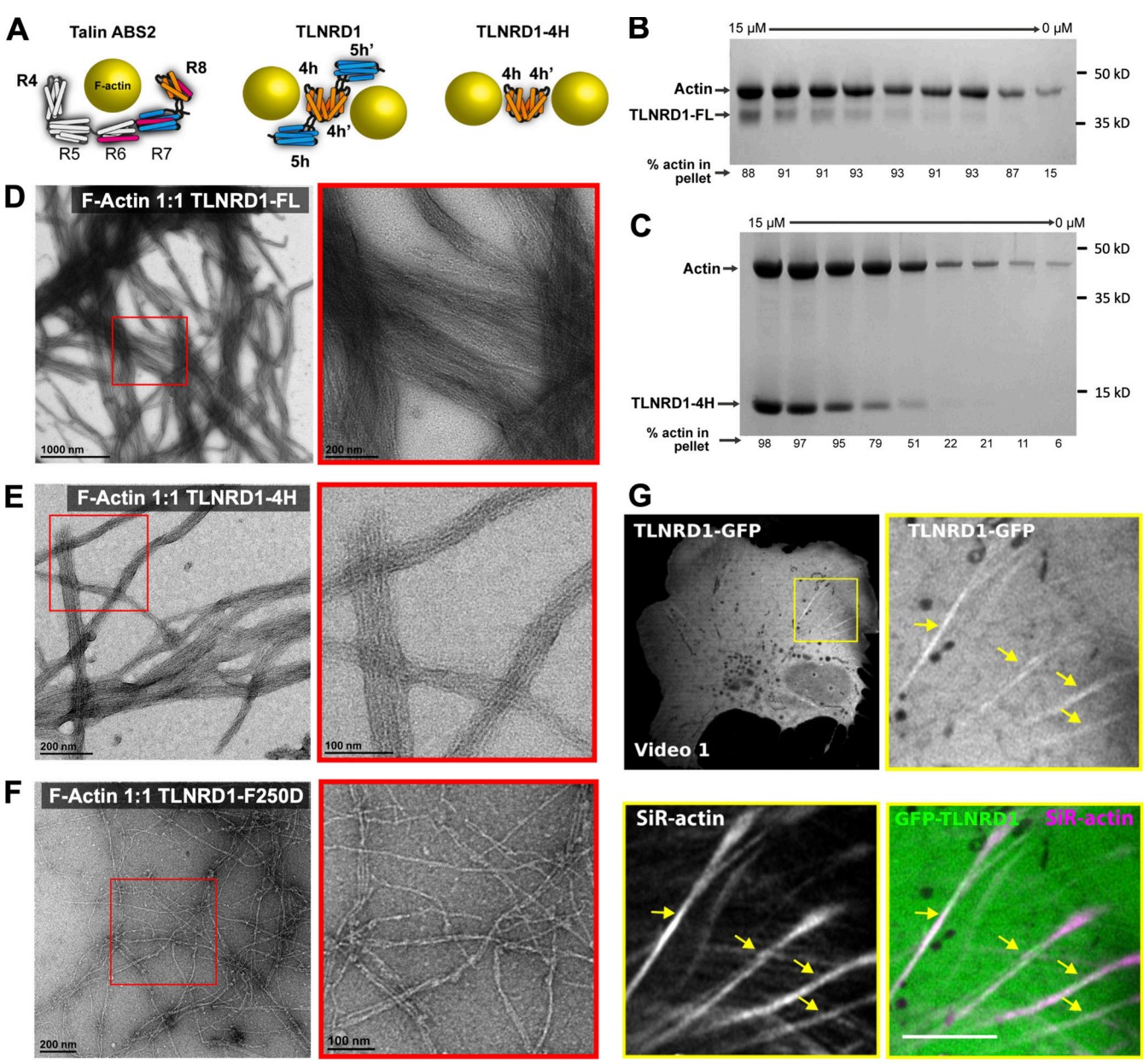

**Figure 4. TLNRD1 is an actin-bundling protein. (A)** Schematic of talin ABS2 binding a single actin filament (left). TLNRD1-FL (middle) and TLNRD1-4H (right) bind two actin filaments. **(B and C)** Actin bundling assays using serial dilutions of TLNRD1-FL (B) and TLNRD1-4H (C) against 15 μM F-actin. **(D–F)** EM images of negative-stained F-actin bundles with (D) TLNRD1-FL, (E) TLNRD1-4H, and (F) TLNRD1-F250D. Red box (left) enlarged on right. **(G)** U2OS cells expressing GFP-TLNRD1 were plated on fibronectin, incubated for 2 h with SiR-actin to label the actin cytoskeleton, and imaged live using an Airyscan confocal microscope (1 picture every 16 s; Video 1). Scale bars: (main) 25 μm, (inset) 5 μm. The yellow arrows highlight GFP-TLNRD1 localizing to actin stress fibers.

accumulates to MYO10 filopodia tips. Altogether, our data demonstrate that TLNRD1 is a filopodia protein that modulates filopodia formation. Finally, the effect of TLNRD1 on cell migration was assessed using live-cell imaging of U2OS cells plated on fibronectin. This revealed that expression of GFP-TLNRD1 significantly increased migration velocity compared with non-transfected, or GFP-expressing, cells (Fig. S3 C).

## Conclusions

TLNRD1 shares high sequence and structural similarity with the R7R8 region of talin, an important signaling hub that coordinates multiple cellular pathways. In this work, we have characterized

TLNRD1 as a novel actin-bundling protein that supports formation of filopodia.

Talin R7R8 has multiple roles, binding to actin and coupling talin to several ligands containing LD motifs. Here we show that TLNRD1 retains the capacity to bind actin and the LD proteins RIAM and KANK. However, it lacks the vinculin binding site (VBS) found in talin R7R8. Given the high similarity between talin R7R8 and TLNRD1, TLNRD1 might act as a dominant-negative modulator of talin function, fine-tuning talin signaling responses by sequestering R7R8 ligands and thus uncoupling them from their connection to integrin-mediated adhesion complexes. We also demonstrate that TLNRD1 is an obligate dimer with a mode of

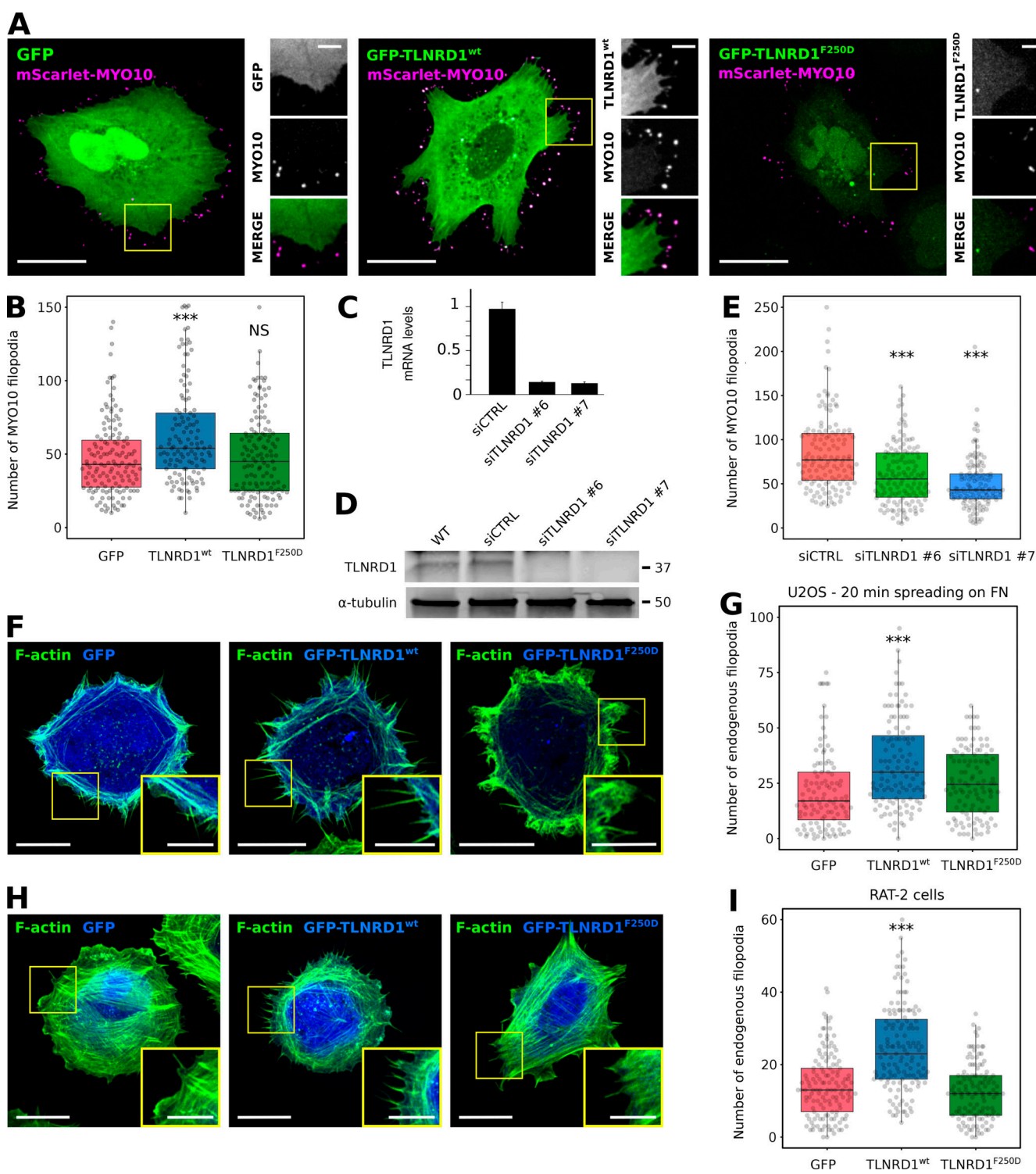

Figure 5. **TLNRD1 localizes to, and modulates, filopodia. (A and B)** U2OS cells transiently expressing MYO10-mScarlet and GFP, GFP-TLNRD1, or GFP-TLNRD1-F250D were plated on fibronectin for 2 h, fixed, and imaged using a spinning disk confocal microscope. Representative images are displayed (A). Yellow square indicates region of interest, which is magnified on the right. Scale bars: (main) 25 µm, (inset) 5 µm. **(B)** The number of MYO10-positive filopodia per cell was then quantified (*n* > 125 cells, three biological repeats; ***, P = 0.002). **(C and D)** Efficiency of siRNA-mediated (oligos nos. 6 and 7) silencing of TLNRD1 in U2OS cells as detected using quantitative PCR (C) or Western blotting (D). To detect TLNRD1 protein levels by Western blot, TLNRD1 was first immuno-precipitated from cell lysate as indicated. **(E)** TLNRD1-silenced (oligos nos. 6 and 7) U2OS cells transiently expressing MYO10-GFP were plated on fibronectin for 2 h and fixed, and the number of MYO10-positive filopodia per cell was quantified (*n* > 150 cells, three biological repeats; ***, P < 0.001). **(F and G)** U2OS cells transiently expressing GFP, GFP-TLNRD1, or GFP-TLNRD1-F250D were plated on fibronectin for 20 min, fixed, and imaged using an Airyscan confocal microscope. Scale bars: (main) 10 µm, (inset) 5 µm. **(G)** The number of endogenous filopodia per cell was quantified (*n* > 108 cells, three biological repeats; ***, P < 0.001). **(H and I)** RAT-2 cells transiently expressing GFP, GFP-TLNRD1, or GFP-TLNRD1-F250D were plated on fibronectin for 2 h,

fixed, and imaged using a spinning disk confocal microscope. Scale bars: (main) 20 μm, (inset) 5 μm. **(I)** The number of endogenous filopodia per cell was quantified ($n > 129$ cells, four biological repeats; ***, $P < 0.001$). P values were determined using a randomization test (see Materials and methods for details). siCTRL, siRNA used as control.

actin binding distinct from talin ABS2. The orientation of the two TLNRD1 monomers with respect to actin allows the dimeric protein to efficiently bundle actin. Together, this suggests that TLNRD1 has similar and distinct functions to talin R7R8.

Talin R7R8 is buried in the autoinhibited form of the molecule (Dedden et al., 2019), and is exposed upon talin activation at adhesion sites. As TLNRD1 lacks these additional domains, it might be constitutively active (Fig. 3 H). However, a notable feature of the TLNRD1-FL structure is that the five-helix bundle packs against the side of the four-helix bundle. We observe a similar compact conformation in solution (Fig. S2, I and J). In contrast, multiple structures of talin1 R7R8 solved alone (Gingras et al., 2010) and in complex (Chang et al., 2014; Zacharchenko et al., 2016; Gough et al., 2021) show no equivalent interactions between R7 and R8. Thus, it is possible that this more compact conformation might represent an autoinhibited conformation of TLNRD1, rendering some binding sites cryptic.

Importantly, we report that TLNRD1 dimers but not monomers localize to and regulate filopodia formation (Fig. 5). As TLNRD1 dimers bundle actin, and as actin bundlers have a well-characterized role in inducing filopodia (Gupton and Gertler, 2007; Khurana and George, 2011; Jacquemet et al., 2015), it is tempting to speculate that TLNRD1 promotes filopodia formation via its actin-bundling property. As filopodia often contribute to invasive migration of cancer cells (Jacquemet et al., 2015, 2017), and as increased TLNRD1 mRNA levels correlate with poor outcomes in lung cancer (Nagy et al., 2018), future work will focus on the contribution of TLNRD1 to invasive cell migration.

In summary, we have structurally and biochemically characterized the TLNRD1 protein as a novel actin-bundling protein that can drive filopodia formation and cell migration. In doing so, we have expanded both the talin family of proteins with the identification of a third talin gene and the growing set of proteins that can polymerize actin filaments into bundles.

## Materials and methods

### Constructs for biochemical/structural assays
Human TLNRD1 (TLNRD1-FL, residues 1–362) pET151 was purchased as a codon-optimized synthetic gene from GeneArt. TLNRD1-4H (residues 143–273) was sub-cloned into pET151 vector. Single point mutations were introduced into TLNRD1 using site-directed mutagenesis with *Pfu* DNA polymerase (Promega), followed by digestion with DpnI at 37°C for 1 h and transformation into DH10β *Escherichia coli* cells.

All expression constructs have been deposited in Addgene at http://www.addgene.org/ben_goult.

### Protein expression and purification
TLNRD1 constructs were expressed in BL21(DE3) *E. coli* cells grown in lysogeny broth at 37°C with 100 μg/ml ampicillin. Expression was induced with 0.1 mM IPTG and cells further incubated at 18°C overnight. Proteins were purified using standard techniques (Khan et al., 2021). Briefly, following centrifugation, pelleted cells were resuspended in nickel affinity buffer (50 mM imidazole, 500 mM NaCl, and 20 mM Tris-HCl, pH 8). Cell lysates were loaded onto a 5-ml HisTrap HP column (GE Healthcare) for purification by nickel affinity chromatography. Eluted protein was exchanged into MES buffer (20 mM 2-N-morpholinoethanesulfonic acid, 20 mM NaCl, and 2 mM DTT, pH 6.5). His-tags were removed with AcTEV protease (Invitrogen) overnight and proteins further purified with a HiTrap SP HP cation exchange column (GE Healthcare).

### Fluorescence polarization assay
The following peptides with a C-terminal cysteine residue were synthesized by GLBiochem (Shanghai): KANK1(30–60) PYF-VETPYGFQLDLDFVKYVDDIQKGNTIKK-C, KANK1(30–60)-4A PYFVETPYGFQAAAAFVKYVDDIQKGNTIKK-C, lamellipodin(20–46) EDQDLDKMFGAWLGELDRLTQSLDSDK-C, and RIAM(4–30) SEDIDQMFSTLLGEMDLLTQSLGVDT-C.

Assays were performed with a serial dilution of protein, with target peptides at a concentration of 100 nM. Peptides were coupled with either a fluorescein or BODIPY-TMR dye (Thermo Fisher Scientific). Fluorescence polarization was measured using a CLARIOstar plate reader (BMGLabTech) at 20°C. Data were analyzed using GraphPad Prism 7 software and $K_d$ values generated with the one-site total binding equation.

### F-actin cosedimentation assays
Actin was isolated from rabbit muscle acetone powder kindly gifted by Professor Mike Geeves (University of Kent, UK) and prepared using a cycle of polymerization and depolymerization following the protocol of Spudich and Watt (1971). Briefly, 1.5 g of acetone powder was stirred on ice in prechilled buffer (10 mM Tris, 1 mM DTT, 0.5 mM ATP, and 0.2 mM $CaCl_2$, pH 8) for 30 min, filtered, and spun at 30,000 rpm for 1 h at 4°C. The actin was polymerized by adding KCl to a final concentration of 100 mM followed by $MgCl_2$ to a final concentration of 2 mM and stirred at room temperature for 1 h. The polymerized actin was then pelleted by centrifugation at 30,000 rpm at 4°C for 3 h before being resuspended in depolymerizing buffer (5 mM Tris, 1 mM $NaN_3$, and 0.2 mM $CaCl_2$, pH 7.5) by homogenization and dialyzed into the same buffer overnight at 4°C. The dialyzed depolymerized actin was centrifuged the next day at 30,000 rpm at 4°C for 1 h to remove sediments and diluted to a concentration of 1 mg/ml. Final purified F-actin was stored at 4°C in polymerization buffer (10 mM Tris-HCl, pH 7, 50 mM NaCl, 2 mM $MgCl_2$, 1 mM $NaN_3$, and 1 mM DTT). For cosedimentation assays, F-actin was diluted to 15 μM and incubated with a serial dilution of protein starting at a 1:1 ratio for 1 h at room temperature. To test binding, samples were spun at 100,000 × $g$ for 20 min at 4°C. To test bundling activity, samples were spun at 10,000 × $g$ for 15 min at 4°C. Equal volumes of pellet and supernatant were

loaded onto SDS-PAGE gels and densities analyzed using ImageJ software (Schneider et al., 2012).

## Negative stain EM
Human F-actin (cat. no. APHL99-E; Cytoskeleton) was diluted to 23 μM in polymerization buffer. TLNRD1-FL, TLNRD1-4H, and TLNRD1-F250D were incubated with F-actin at a 1:1 ratio overnight at 4°C. After incubation, samples were diluted down to 1 μM with polymerization buffer. Samples were applied to 400 mesh carbon-coated copper grids for 30 s and negatively stained for 30 s with 2% (wt/vol) uranyl acetate. Excess stain was removed and grids air-dried. Images were taken on an FEI Tecnai T12 EM equipped with a Gatan US4000 CCD detector and accelerating voltage of 120 kV.

## SEC-MALS
SEC-MALS analysis of TLNRD1-FL and TLNRD1-F250D mutant was performed at room temperature with 100 μl of protein at 150 μM. Samples were loaded onto a Superdex 75 size-exclusion column (GE Healthcare Life Sciences) and eluted proteins measured by Viscotek Sec-Mals 9 and Viscotek R1 detector VE3580 (Malvern Panalytical). Data were analyzed using OmniSEC software.

## Crystallization, x-ray data collection, and structure solution
TLNRD1-FL and TLNRD1-4H crystallization trials were performed using hanging drop vapor diffusion at 21°C with concentrations of 390 μM and 350 μM, respectively. Crystals of TLNRD1-FL were obtained in a condition containing 250 mM NaSCN and 20% PEG3350 and grown to optimal size in 7 d. For TLNRD1-4H, crystals were obtained in 300 mM HOC(CO$_2$H)(CH$_2$CO$_2$NH$_4$)$_2$ and 25% PEG3350 and attained full growth in 4 d. Crystals were harvested in their respective growth solutions supplemented with 25% ethylene glycol as cryoprotectant, mounted on CryoLoops (Hampton Research) or LithoLoops (Molecular Dimensions), and vitrified in liquid nitrogen for data collection. x-ray diffraction datasets were collected at 100 K at Proxima-1 beamline at Soleil Synchrotron using a Pilatus 6M detector (Dectris) and processed by an autoPROC pipeline (Vonrhein et al., 2011), which incorporates XDS (Kabsch, 2010), AIMLESS (Evans and Murshudov, 2013), and TRUNCATE (Evans, 2011) for data integration, scaling, and merging, respectively. The structure of TLNRD1-4H was determined by molecular replacement performed by PHASER (McCoy et al., 2007) using Protein Data Bank accession no. 2X0C (Gingras et al., 2010) as a search model. For the TLNRD1-FL solution, BALBES molecular replacement pipeline (Long et al., 2008) was employed to generate the initial model, which was then manually tweaked before adjustment and refinement. Manual model adjustment was performed in COOT (Emsley et al., 2010) followed by refinement using PHENIX.REFINE (Afonine et al., 2012). TLNRD1-FL (Protein Data Bank accession no. 6XZ4) was diffracted to 2.3 Å in $P2_1$ space group with two TLNRD1 molecules in the asymmetric unit, and TLNRD1-4H (Protein Data Bank accession no. 6XZ3) to 2.2 Å in $I4_122$ space group with one molecule in the asymmetric unit (statistics in Table S1). For TLNRD1-FL, electron density could be traced for residues 40–341, and for TLNRD-4H, residues 148–270. Interaction properties of the dimer interface of the TLNRD1-FL were assessed by PISA (Krissinel and Henrick, 2007), and figures were prepared in PyMOL (Schrödinger LLC). Models were validated by MOL-PROBITY (Chen et al., 2010) before deposition.

## SEC-small angle x-ray scattering (SAXS)
SEC-SAXS data were collected at Diamond Light Source beamline B21 (Didcot). TLNRD1-FL SAXS experiments were performed at 80 μM and 185 μM in 20 mM Tris, pH 7, 50 mM NaCl, and 2 mM DTT. All samples were loaded onto a KW-403-4F 10–600 kD size-exclusion column (Shodex) connected to an Agilent 1200 HPLC system. Data were analyzed using ScÅtter software available from http://www.bioisis.net and ATSAS online services (Franke et al., 2017).

## Microscale thermophoresis
For investigations into dimerization, His-tagged TLNRD1 was diluted to 100 nM and coupled with His-tag NT-647 dye (RED-tris-NTA; NanoTemper) at room temperature for 30 min. Unlabeled non–His-tagged TLNRD1 was diluted down to 5 μM. Final working concentrations of labeled and unlabeled protein were 50 nM and 2.5 μM, respectively, with the unlabeled protein serially diluted. Samples were loaded into Monolith NT.115 Capillaries and run on a Monolith NT.115 (NanoTemper). All experiments were run at 25°C with 40% laser excitation. Data were analyzed using MO.Affinity Analysis v2.3.

## Cell culture
For cell culture experiments, N-terminal GFP-tagged mouse TLNRD1 was used as previously described (Gingras et al., 2010). The mScarlet-MYO10 construct was described previously (Jacquemet et al., 2019). Human U2OS osteosarcoma cells (Leibniz Institute DSMZ-German Collection of Microorganisms and Cell Cultures) and RAT-2 cells (CRL-1764; American Type Culture Collection) were grown in DMEM supplemented with 10% FCS, 2 mM L-glutamine, and 1% (vol/vol) penicillin and streptomycin, and maintained at 37°C in a humidified 5% CO$_2$ environment.

## Transfection and siRNA knockdown
Plasmids of interest were transfected using Lipofectamine 3000 and the P3000TM Enhancer Reagent (Thermo Fisher Scientific) according to the manufacturer's instructions. The expression of proteins of interest was suppressed using 100 nM siRNA and Lipofectamine 3000 (Thermo Fisher Scientific) according to the manufacturer's instructions. The siRNA used as control was Allstars negative control siRNA (cat. no. 1027281; QIAGEN). The siRNAs targeting TLNRD1 were purchased from QIAGEN (siTLNRD1#6, Hs_MESDC1_6 FlexiTube siRNA, cat. no. SI04217605; siTLNRD1 #7, Hs_MESDC1_7 FlexiTube siRNA, cat. no. SI04314569; and siTLNRD1#8, Hs_MESDC1_8 FlexiTube siRNA, cat. no. SI04362820).

## TLNRD1 immunoprecipitation
For immunoprecipitation experiments, U2OS cells were grown to 90% confluency in a 100-mm dish and lysed with equal volumes of appropriate buffer (40 mM Hepes-NaOH, pH 7.4, 75 mM NaCl, 2 mM EDTA, and 2% NP-40). Lysates were cleaned by centrifugation at 10,000 × $g$ for 5 min at 4°C before 3 h incubation

at 4°C with Dynabeads Protein G superparamagnetic beads precoated with anti-TLNRD1 antibody or IgG control. Beads were washed three times with PBS. Protein extracts were separated under denaturing conditions by SDS-PAGE and subjected to Western blotting with appropriate primary antibody diluted 1:1,000 followed by incubation with the appropriate fluorophore-conjugated secondary antibody diluted 1:5,000. Membranes were scanned using an Odyssey infrared imaging system (LI-COR Biosciences).

Anti-TLNRD1 antibodies were raised in rabbit against recombinantly expressed human TLNRD1 (residues 1–362) by Capra Science. The secondary antibody used for Western blot was an IRDye 800CW conjugated donkey anti-rabbit antibody (cat. no. 926–32213; Li-Cor).

## Sample preparation for light microscopy
For structured illumination microscope (SIM) imaging, U2OS cells transiently expressing GFP-TLNRD1 and Myosin-X-mScarlet were plated on high-tolerance glass-bottom dishes (coverslip no. 1.7; MatTek Corporation) precoated first with poly-L-lysine (10 mg/ml, 1 h at 37°C) and then with bovine plasma fibronectin (10 mg/ml, 2 h at 37°C). After 2 h, samples were fixed and permeabilized simultaneously using a solution of 4% (wt/vol) PFA and 0.25% (vol/vol) Triton X-100 for 10 min. Cells were then washed with PBS, quenched using a solution of 1 M glycine for 30 min, and incubated with silicon rhodamine (SiR)-actin (100 nM in PBS; cat. no. CY-SC001; Cytoskeleton) at 4°C until imaging (minimum length of staining, overnight at 4°C; maximum length, 1 wk). Just before imaging, samples were washed three times in PBS and mounted in Vectashield (Vectorlabs).

To map the localization of each protein within filopodia, images were first processed in Fiji (Schindelin et al., 2012) and data analyzed using R as previously described (Jacquemet et al., 2019). Briefly, in Fiji, the brightness and contrast of each image were automatically adjusted using, as an upper maximum, the brightest cellular structure labeled in the field of view. In Fiji, line intensity profiles (1 pixel width) were manually drawn from filopodium tip to base (defined by the intersection of the filopodium and the lamellipodium). To avoid any bias in the analysis, the intensity profile lines were drawn from a merged image. All visible filopodia in each image were analyzed and exported for further analysis (export was performed using the "Multi Plot" function). For each staining, line intensity profiles were then compiled and analyzed in R. To homogenize filopodia length, each line intensity profile was binned into 40 bins (using the median value of pixels in each bin and the R function "tapply"). Using the line intensity profiles, the percentage of filopodia positive for each protein of interest was quantified. The map of each protein of interest was created by averaging hundreds of binned intensity profiles.

For the filopodia formation assays, cells were plated on fibronectin-coated glass-bottom dishes (MatTek Corporation) as indicated. Samples were fixed for 10 min using a solution of 4% (wt/vol) PFA, then permeabilized using a solution of 0.25% (vol/vol) Triton X-100 for 3 min. Cells were then washed with PBS and quenched using a solution of 1 M glycine for 30 min. Samples were then washed three times in PBS and stored in PBS containing

SiR-actin (100 nM; cat. no. CY-SC001; Cytoskeleton) at 4°C until imaging. Just before imaging, samples were washed three times in PBS. Images were acquired using either a spinning disk confocal microscope (100× objective) or an Airyscan confocal microscope (63× objective). Samples were kept in PBS and imaged at room temperature. The number of filopodia per cell and their length were manually counted using Fiji (Schindelin et al., 2012).

## Microscopy setup
The spinning disk confocal microscope used was a Marianas spinning disk imaging system with a Yokogawa CSU-W1 scanning unit on an inverted Zeiss Axio Observer Z1 microscope controlled by SlideBook 6 (Intelligent Imaging Innovations, Inc.). Images were acquired using a Photometrics Evolve, a back-illuminated EMCCD camera (512 × 512 pixels), and a 100× (NA 1.4 oil, Plan-Apochromat, M27) objective.

The confocal microscope used was a laser scanning confocal microscope LSM880 (Zeiss) equipped with an Airyscan detector (Carl Zeiss) and a 40× oil (1.2) or 63× oil (1.4) objective. The microscope was controlled using Zen Black (2.3), and the Airyscan was used in standard super-resolution mode. Fixed samples were kept in PBS and imaged at room temperature. Live samples were kept in their growing media supplemented by 50 mM of Hepes and imaged at 37°C in the presence of $CO_2$.

The SIM used was DeltaVision OMX v4 (GE Healthcare Life Sciences) fitted with a 60× Plan-Apochromat objective lens, 1.42 NA (immersion oil refractive index of 1.516) used in SIM illumination mode (five phases × three rotations). Emitted light was collected on a front-illuminated pco.edge sCMOS (pixel size 6.5 mm, readout speed 95 MHz; PCO AG) controlled by SoftWorx. Samples were mounted in Vectashield (Vectorlabs) and imaged at room temperature.

## 2D random migration assay
Cells were seeded at a density of $5 \times 10^3$ in 1 ml media supplemented with 50 mM Hepes on plates coated with 10 µg/ml fibronectin and incubated for 2 h at 37°C, 5% $CO_2$. Live cell imaging was performed on a Nikon Eclipse Ti2-E widefield microscope with a heated $CO_2$ chamber, Hamamatsu scientific CMOS Orca Flash 4 v4, and 10× Nikon CFI Plan Fluor objective. Random migration of cells was measured over 24 h in a time-lapse video with images taken every 10 min. Manual tracking of cells was performed using Fiji ImageJ MTrackJ plugin. The tracked data were analyzed using Ibidi chemotaxis and migration tool to determine migration speeds, directionality, and distance. Graphs of resulting data were produced using PlotsOfData (Postma and Goedhart, 2019).

## Quantification and statistical analysis
Randomization tests were performed using the online tool PlotsOf-Differences (https://huygens.science.uva.nl/PlotsOfDifferences/; Goedhart, 2019). Dot plots were generated using PlotsOfData (Postma and Goedhart, 2019).

## Quantitative RT-PCR
Total RNA extracted using the NucleoSpin RNA Kit (Macherey-Nagel) was reverse-transcribed into cDNA using the high-capacity cDNA reverse transcription kit (Applied Biosystems) according to the manufacturer's instructions. The RT-PCR reactions

were performed using predesigned single tube TaqMan gene expression assays and were analyzed with the 7900HT fast RT-PCR System (Applied Biosystems). Data were studied using RQ Manager Software (Applied Biosystems). TLNRD1 primers were from Thermo Fisher Scientific (cat. no. 4351372; probe ID Ss06942862_s1).

## Online supplemental material

Fig. S1 shows the conservation of TLNRD1 and its alignment with talin R7R8 domains. Fig. S2 shows biochemical analysis of TLNRD1 and the characterization of the TLNRD1-F250D mutant. Fig. S3 shows the sub-filopodial localization of TLNRD1 in MYO10 filopodia and the role of TLNRD1 in migrating cells. Video 1 shows that TLNRD1 can localize to actin fibers in living cells. Table S1 lists the crystallographic data collection and refinement statistics for TLNRD1-FL and TLNRD1-4H.

## Acknowledgments

We thank David Critchley for critical reading of the manuscript and Anthony Baines for his help with the bioinformatics of the *TLNRD1* gene. We also thank the Soleil synchrotron beamline Proxima-1 staff for their help in crystallographic data collection. We thank J. Siivonen and P. Laasola for technical assistance and M. Saari for help with the microscopes. The Cell Imaging and Cytometry Core facility (Turku Bioscience, University of Turku, Åbo Akademi University and Biocenter, Finland) is acknowledged for services, instrumentation, and expertise.

B.T. Goult was funded by Biotechnology and Biological Sciences Research Council grants (BB/N007336/1 and BB/S007245/1), and B.T. Goult and A. Akhmanova were funded by Human Frontier Science Program grant (RGP00001/2016). G. Jacquemet was supported by grants awarded by the Academy of Finland, the Sigrid Juselius Foundation, and the Åbo Akademi University Research Foundation (CoE CellMech), and by Drug Discovery and Diagnostics strategic funding to Åbo Akademi University.

The authors declare no competing financial interests.

Author contributions: A.R. Cowell and B.T. Goult conceptualized the study. A.R. Cowell, G. Jacquemet, A.K. Singh, L. Varela, A.S. Nylund, and Y-C. Ammon performed experiments. A.R. Cowell, G. Jacquemet, A.K. Singh, L. Varela, A.S. Nylund, Y-C. Ammon, D.G. Brown. A. Akhmanova, J. Ivaska, and B.T. Goult analyzed data. A.R. Cowell, G. Jacquemet, A. Akhmanova, J. Ivaska, and B.T. Goult wrote and edited the manuscript.

Submitted: 29 May 2020

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

# Supplemental material

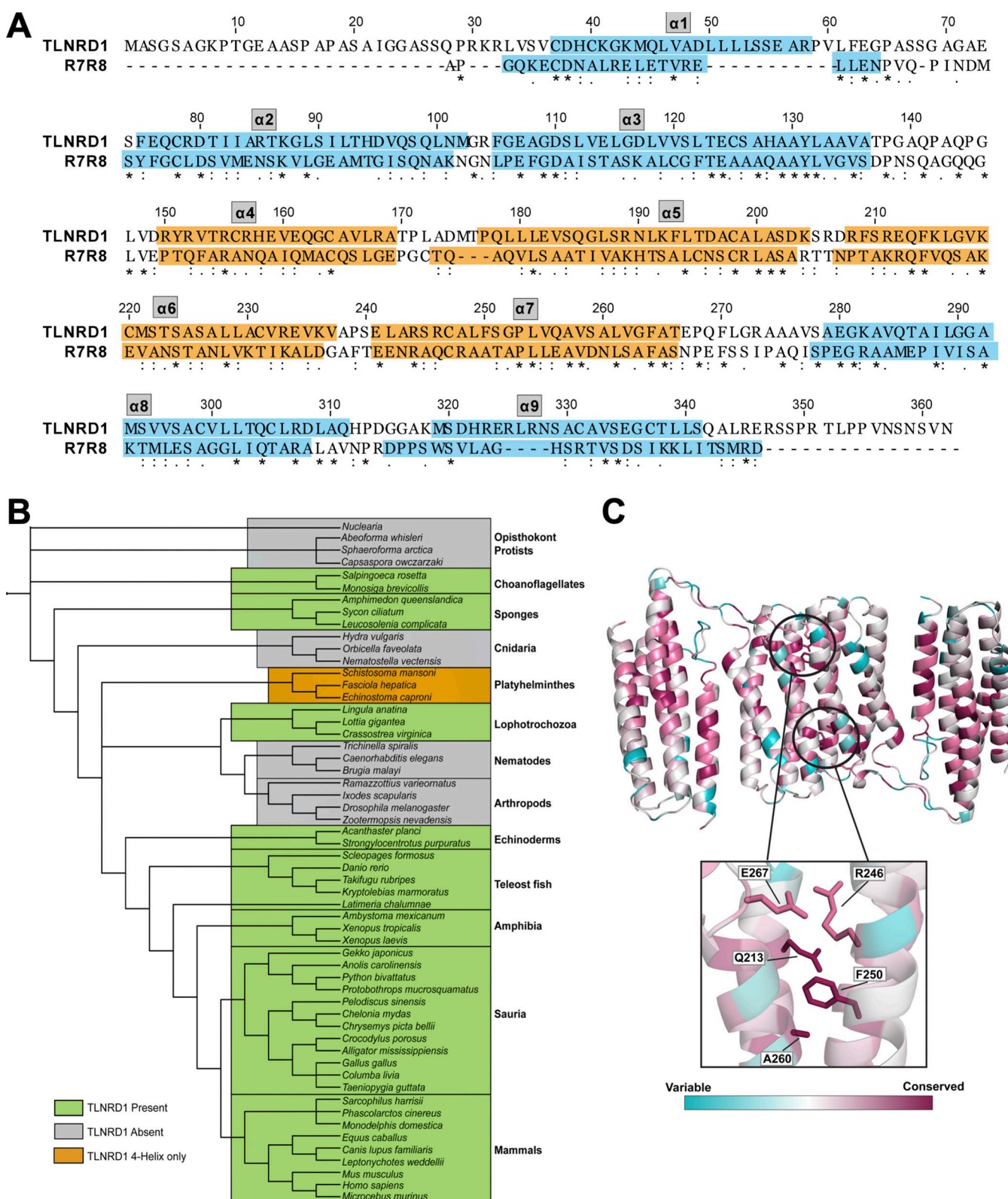

Figure S1. **Conservation of TLNRD1 and its alignment with talin R7R8 domains. (A)** Amino acid sequence alignment of human TLNRD1 (UniProt accession no. Q9H1K6) and talin1 residues 1357–1653 encompassing R7R8 (UniProt accession no. Q9Y490). Domain boundaries are highlighted with orange corresponding to the four-helix domain and blue corresponding to the five-helix domain. **(B)** Tree diagram representing TLNRD1 presence and absence over evolution. TLNRD1 first appears in choanoflagellates and sponges and is conserved throughout vertebrate evolution. TLNRD1 has been lost from nematodes, arthropods, and Cnidaria. **(C)** TLNRD1-FL sequence conservation mapped onto the full-length TLNRD1 structure using ConSurf (Ashkenazy et al., 2016). Structure colored according to extent of conservation with blue indicating variability and purple indicating highly conserved residues. Residues important for dimerization (Q213, R246, F250, A260, and E267) are highlighted.

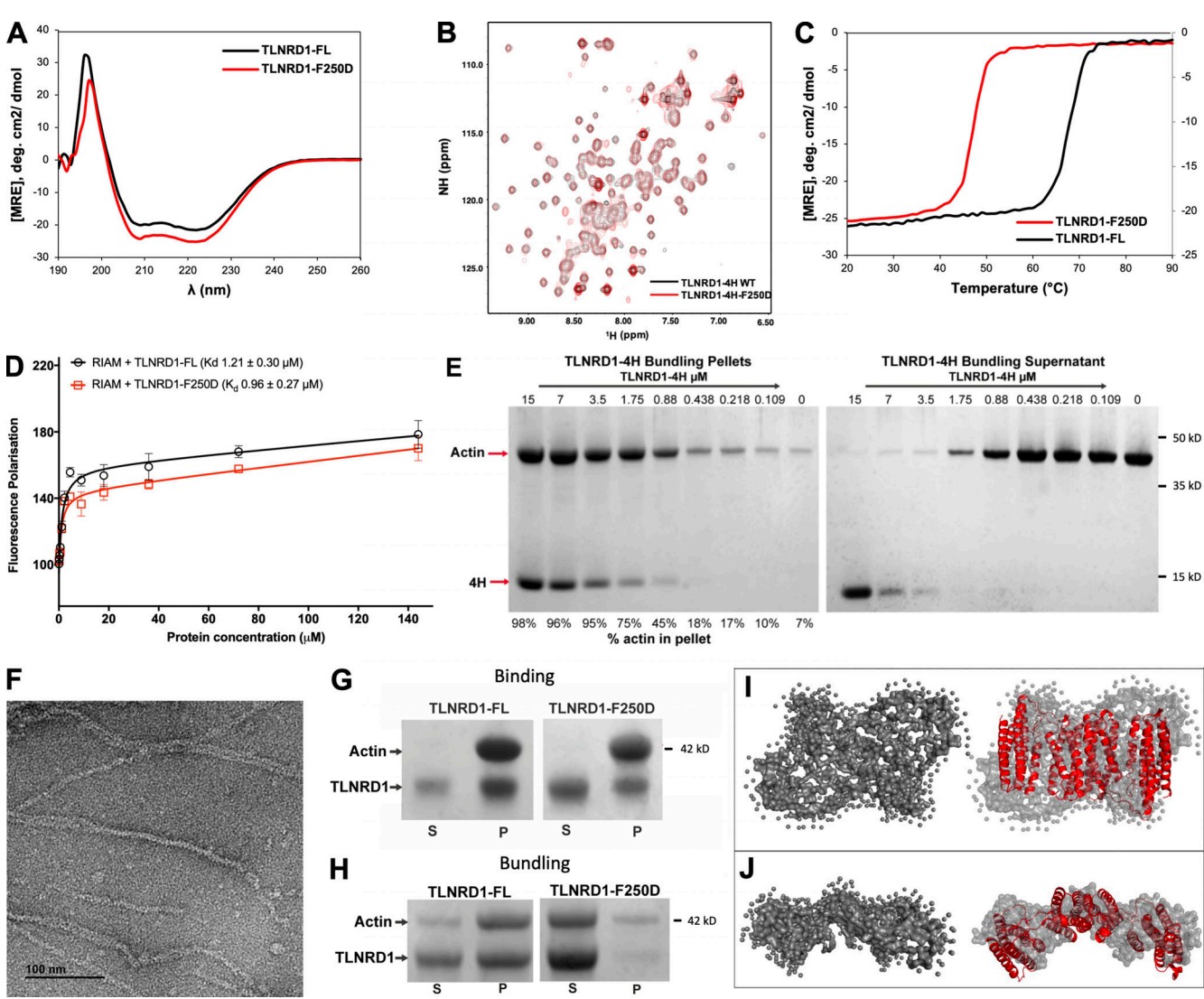

Figure S2. **Biochemical analysis of TLNRD1. (A)** CD spectra of TLNRD1-FL (black) and TLNRD1-F250D (red) showing no change in overall secondary structure as a result of the mutation. **(B)** nNMR analysis of the F250D mutant shows it closely resembles the WT. $^1H,^{15}N$ heteronuclear single quantum coherence (HSQC) spectra of 150 µM $^{15}N$-labeled TLNRD1–4H WT (black) or TLNRD1-4H-F250D (red). **(C)** CD thermostability analysis of TLNRD1-FL and TLNRD1-F250D showing a reduction in stability with loss of dimerization. **(D)** Binding of fluorescein-labeled RIAM (residues 4–30) peptide with TLNRD1-FL WT (black) and TLNRD1-FL F250D (red) measured using a fluorescence polarization assay. **(E)** Low-speed actin cosedimentation bundling assay with TLNRD1-4H serial dilution against 15 µM F-actin. The two gels show the (left) pellet and (right) supernatant fractions. At high TLNRD1-4H concentrations, unbound TLNRD1-4H is present in the supernatant, suggesting the binding becomes saturated. At low TLNRD1-4H concentrations, the actin is predominantly in the supernatant. **(F)** Actin filaments alone visualized by negative stain EM. **(G and H)** Actin binding of the F250D mutant. **(G)** High-speed actin binding assay of TLNRD1-FL and TLNRD1-F250D showing little impact on actin binding with loss of dimerization. **(H)** Low-speed actin cosedimentation assay showing loss of TLNRD1 actin bundling with loss of dimerization as a result of the F250D mutation. **(I and J)** SAXS analysis of TLNRD1 shows compact domain arrangement in solution. **(I)** SAXS envelope reconstruction with GASBOR, showing the best fit with the TLNRD1-FL crystal structure. **(J)** Top-down view of I. MRE, molar residue ellipticity; deg., degrees; NH, amide.

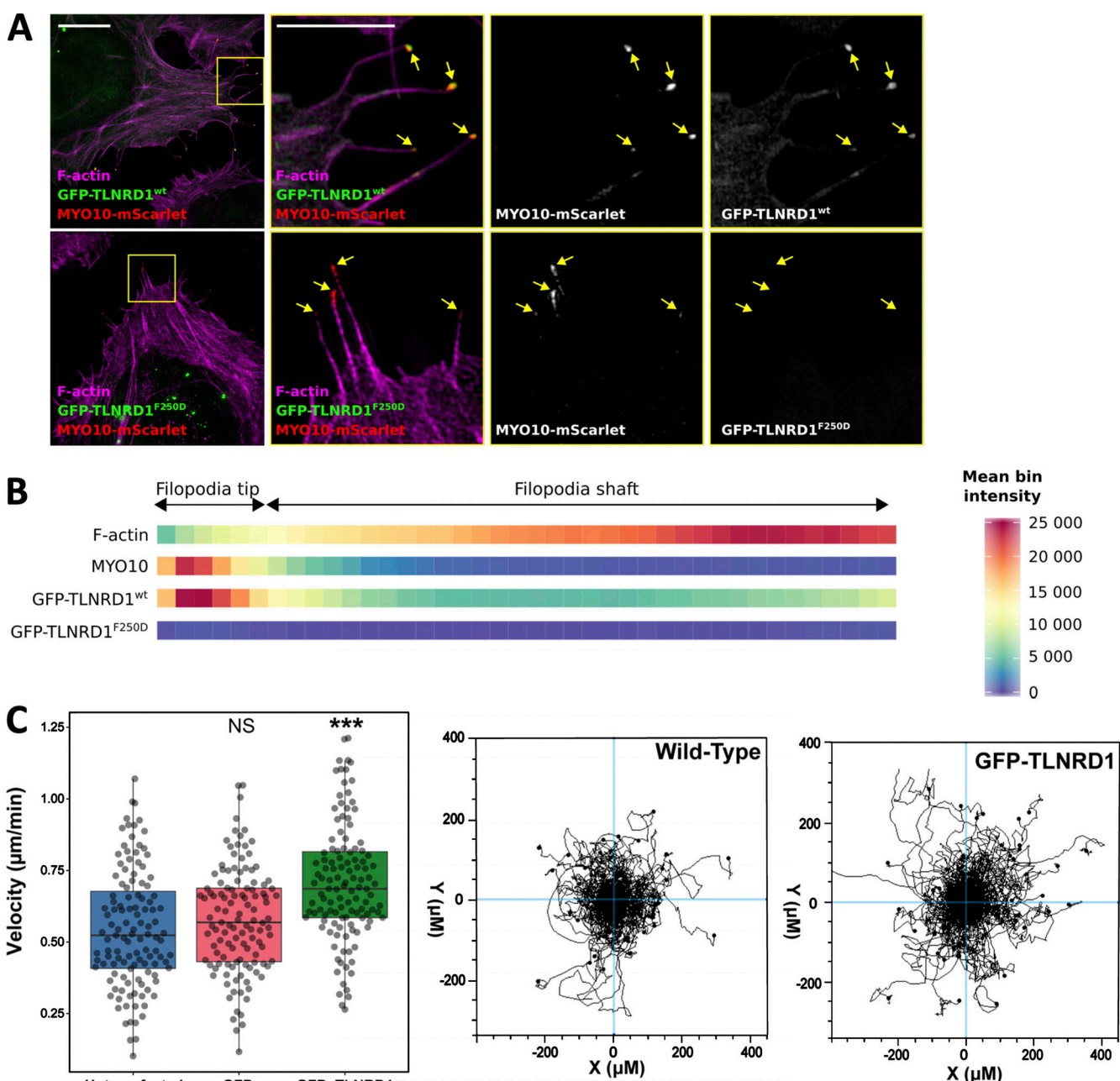

Figure S3. **TLNRD1 localizes to the tip of MYO10 filopodia.** U2OS cells expressing MYO10-mScarlet and GFP-TLNRD1 or GFP-TLNRD1-F250D were plated on fibronectin for 2 h, stained for F-actin, and imaged using SIM. **(A)** Representative maximum intensity projections are displayed. The yellow squares highlight regions of interest, which are magnified; scale bars: (main) 10 μm; (inset) 5 μm. **(B)** Heatmap highlighting the subcellular localization of F-actin, MYO10, TLNRD1, and TLNRD1-F250D within filopodia based on >360 intensity profiles (see Materials and methods for details). **(C)** Random 2D migration assay of U2OS cells plated on fibronectin and nontransfected or transiently expressing GFP or GFP-TLNRD1. GFP-TLNRD1 overexpression increases migration velocity in 2D ($n$ = 120 cells from three independent repeats; ***, P < 0.001). Cell trajectories for nontransfected and GFP-TLNRD1–expressing cells are shown. P values were determined using a randomization test (see Materials and methods for details).

Video 1. **TLNRD1 can localize to actin fibers in living cells.** U2OS cells expressing GFP-TLNRD1 were plated on fibronectin, incubated for 2 h with SiR-actin to label the actin cytoskeleton, and imaged live using an Airyscan confocal microscope (1 picture every 16 s).

**Table S1 is provided online as a separate file. Table S1 shows data collection and refinement statistics for TLNRD1-FL and TLNRD1-4H domains.**

