## [Peer Review File · The Journal of Cell Biology]

Talin Rod Domain-Containing 1 (TLNRD1) is a novel actin-bundling protein which promotes filopodia

Alana Cowell, Guillaume Jacquemet, Abhimanyu Singh, Lorena Varela, Anna Nylund, York-Christoph Ammon, David Brown, Anna Akhmanova, Johanna Ivaska, and Benjamin Goult

Corresponding Author(s): Benjamin Goult, University of Kent

Review Timeline:

Submission Date:	2020-05-29
Editorial Decision:	2020-06-25
Revision Received:	2021-06-03
Editorial Decision:	2021-06-10
Revision Received:	2021-06-17

Monitoring Editor: Kenneth Yamada

Scientific Editor: Tim Spencer

Transaction Report:

DOI: <https://doi.org/10.1083/jcb.202005214>

June 25, 2020

Re: JCB manuscript #202005214

Dr. Benjamin Thomas Goult
University of Kent
School of Biosciences,
14/21 Ingram Building
Canterbury, Kent CT2 7NJ
United Kingdom

Dear Dr. Goult,

Thank you for submitting your manuscript entitled "Talin Rod Domain-Containing 1 is a novel actin-bundling protein which promotes filopodia formation". The manuscript was assessed by expert reviewers, whose comments are appended to this letter. In the reviews from three well-established experts in fields that overlapped the findings reported in this manuscript, there were differences of opinion about the priority of this study for JCB expressed in the comments and recommendations for the Editors. However, our overall conclusion based on the specific comments for authors is that this study could be potentially appropriate for publication in JCB if appropriately revised. Consequently, we invite you to submit a revision if you can address the reviewers' key concerns, as outlined here.

A particularly significant question is whether the findings with talin R7-R8 are valid for full-length talin. From an editorial point of view, establishing similarity to full-length in accessibility of sites and function could alleviate the concerns raised by two reviewers to the Editors about its level of interest. The problem that mutagenesis at a single amino acid residue affects thermal stability is a concern, so additional evidence would considerably strengthen the conclusions. In addition, controls and stronger quantitative evidence would be needed. Moreover, do you have evidence that TLNRD1 affects filopodia in non-over-expressors?

We strongly encourage you to make every effort to resolve the concerns of these conscientious reviewers. While you are revising your manuscript, please also attend to the following editorial points to help expedite the publication of your manuscript. Please direct any editorial questions to the journal office.

GENERAL GUIDELINES:

Text limits: Character count for a Report is < 20,000, not including spaces. Count includes title page, abstract, introduction, results, discussion, acknowledgments, and figure legends. Count does not include materials and methods, references, tables, or supplemental legends.

Figures: Reports may have up to 5 main text figures. To avoid delays in production, figures must be prepared according to the policies outlined in our Instructions to Authors, under Data Presentation, <http://jcb.rupress.org/site/misc/ifora.xhtml>. All figures in accepted manuscripts will be screened prior to publication.

***IMPORTANT: It is JCB policy that if requested, original data images must be made available.

Failure to provide original images upon request will result in unavoidable delays in publication. Please ensure that you have access to all original microscopy and blot data images before submitting your revision.***

Supplemental information: There are strict limits on the allowable amount of supplemental data. Reports may have up to 3 supplemental figures. Up to 10 supplemental videos or flash animations are allowed. A summary of all supplemental material should appear at the end of the Materials and methods section.

As you may know, the typical timeframe for revisions is three to four months. However, we at JCB realize that the implementation of social distancing and shelter in place measures that limit spread of COVID-19 also pose challenges to scientific researchers. Lab closures especially are preventing scientists from conducting experiments to further their research. Therefore, JCB has waived the revision time limit. We recommend that you reach out to the editors once your lab has reopened to decide on an appropriate time frame for resubmission. Please note that papers are generally considered through only one revision cycle, so any revised manuscript will likely be either accepted or rejected.

Thank you for this interesting contribution to the Journal of Cell Biology. You can contact us at the journal office with any questions, cellbio@rockefeller.edu or call (212) 327-8588.

With kind regards,

Ken Yamada
Kenneth Yamada, MD, PhD
Editor, Journal of Cell Biology

Melina Casadio, PhD
Senior Scientific Editor, Journal of Cell Biology

Reviewer #1 (Comments to the Authors (Required)):

This an elegant study describing the experimental path from a detailed structural characterization of the TLNRD1 protein, unveiling its constitutive dimerization mechanism and the link to the actin-bundling activity, accomplished by the finding that TLNRD1 is a filopodia tip protein capable of modulating filopodia formation. However, I have identified several issues that should be addressed.

1) The authors tried to obtain the Kd value for dimerization using MST. Due to the experimental setup, the value is significantly underestimated, as for the wild type protein they were titrating the labeled protein by unlabeled material that was already dimeric. Therefore, there were additional equilibria involved. Perhaps AUC or FP would be more appropriate.

2) The low speed cosedimentation assay data (Figure 4B/C) are insufficiently annotated: it is not clearly stated that we are looking at pelleted actin; band intensities do not correspond to % below, and there is a different actin concentration from that stated in methods. Perhaps, more careful quantification/data representation (graphs) might point at a role of 5H in actin binding. The SDS-PAGE data can be then moved to SI.

3) The actin cosedimentation bundling assay was done for both FL and 4H constructs, but the visualization (Figure 4D/E) was limited only to the FL protein. Obtaining analogous data for at least negative-stain EM might provide a better insight into the role of the 4H core in actin bundling.

Reviewer #2 (Comments to the Authors (Required)):

The report by Cowell et al., describes biochemistry and structural characterization of TLNRD1, accompanied by basic cell biological assays.

TLNRD1 is a 2 helical-bundle containing protein that has a high similarity to the R7-R8 domain of the focal adhesion protein talin.

The authors found that TLNRD1 dimerizes through the 2nd module helix bundle, particularly through hydrophobic interaction of F250. In addition, they observed actin bundling, which depends on F250-mediated TLNRD1 dimerization.

Cell biological experiments validated the relevance of residue F250 and TLNRD1's filopodia promotion.

Overall, the results of the study are robust and the experiments are carried out thoroughly.

My concerns are the following:

1. The side-by-side comparison to the function of talin R7-R8 should also extend to full-length talin. Recently Dedden et al. (2019), reported that R7R8 is inhibited from actin binding in its autoinhibited form. Therefore, it is important to state that R7-R8 may function differently in the full length context compared to a truncated version. The 3D architecture of talin is relevant for the recruitment process and also impacts the function and accessibility of R7-R8. This has an implication for talin R7R8, as it would not act as a dimerization module when incorporated in the structural core of talin full-length, while TLNRD1 can work as an independent isolated module.

2. I am curious if talin R7R8 would make an interaction with TLNRD1, possibly to form a heterodimer?

3. The discussion is too lengthy for the contents of this report and should be more concise. The authors give too many speculations for the presented data. Aggressive cancer cell formation seems out of place. The study describes the function of filopodia formation and does not go beyond localization.

4. In the introduction the authors stated TLNRD1 may impact talin R7-R8 binding partners and therefore talin's function as well. Although the biochemical basis of TLNRD1 interacting with R7-R8 binding proteins is shown, cellular experiments are not convincing enough to support this notion. It should be removed from the introduction or extended by further experimental evidence.

Reviewer #3 (Comments to the Authors (Required)):

This manuscript investigates the relatively poorly studied protein TLNRD1, confirming its expected structural organization, dimerization, F-actin binding and interactions with other talin-binding partners but also revealing an unanticipated mode of dimerization and an impact on myosin X stimulated filopodia formation. Data show that TLNRD1 adopted the predicted 5+4 helical bundle architecture but notably also dimerizes via the 4-helical bundle and this appears to support F-actin bundling. TLNRD1 localizes at stress fibers and in cells stimulated to produce filopodia through over-expression of myosin X it localizes at filopodia tips. Filopodia localization and the ability of TLNRD1 to stimulate filopodia formation seems to depend on its dimerization although this conclusion rests on use of a single point mutant that also reduces thermal stability. It is unclear whether TLNRD1 is bundling actin at filopodia tips or is recruited there by other interactions.

The authors report crystal structures of both domains of TLNRD1 or of the 4 helical bundle in isolation. The isolated 4-helix bundle structure is not clearly presented in the manuscript but seems to be very similar to the longer structure and to dimerize in a similar manner. The dimerization mode is unexpected and is supported by mutagenesis at one residue but this dimerization disrupting mutation also leads to a dramatic reduction in thermal stability - this is acknowledged but care is needed in ascribing the functional consequences of this mutation to a lack of dimerization rather than a lack of stable folding. If other residues along the dimer interface are mutated do they have similar effects on both thermal stability and dimerization, or can they separate the two properties?

The authors use FP assays to show that TLNRD1 retains the binding activities of talinR7R8. These data would be enhanced by a direct comparison of the TLNRD1 and talinR7R8 binding results - are affinities comparable? In addition, the authors should comment on the availability of the binding sites in the TLNRD1 structure based on comparison with the talin complexes.

The F-actin co-sedimentation data in Fig 3G lacks controls. This is particularly relevant for the 4H construct which exhibited unanticipated actin binding. Negative controls that do not co-precipitate are needed, as is evidence that the protein is not present in the pellet in the absence of F-actin. Again, a direct parallel comparison with the R7R8 protein might be helpful. Co-sedimentation assays can be used to estimate affinity of the interaction with F-actin and this should be performed, allowing comparison of the wild-type and F250D mutants to ensure that the defective bundling does not relate to reduced affinity for F-actin.

The importance of dimerization for bundling is only shown in the TEM image, the quantitative assays (Fig 4B&C) should be repeated to more clearly show the importance of dimerization. As already mentioned, testing additional mutations along the dimerization interface would also be helpful. Identification of mutations that prevent F-actin binding would also be very helpful but may lie beyond the scope of this work.

The observation that TLNRD1 influences filopodia is interesting but as this is apparently always assessed in MYO10 over-expressing cells its general significance is hard to determine.

Detailed point by point response to reviewer comments:

Reviewer #1 (Comments to the Authors):

“This an elegant study describing the experimental path from a detailed structural characterization of the TLNRD1 protein, unveiling its constitutive dimerization mechanism and the link to the actin-bundling activity, accomplished by the finding that TLNRD1 is a filopodia tip protein capable of modulating filopodia formation. However, I have identified several issues that should be addressed.”

We thank the reviewer for the positive evaluation of the study and for the useful comments that have improved the manuscript.

1) “The authors tried to obtain the K_d value for dimerization using MST. Due to the experimental setup, the value is significantly underestimated, as for the wild type protein they were titrating the labeled protein by unlabeled material that was already dimeric. Therefore, there were additional equilibria involved. Perhaps AUC or FP would be more appropriate.”

Thank you for this suggestion. The use of MST to measure dimerization is well established (Lin et al., 2012; Seidel et al., 2013), and we now reference these prior examples. The correct term for the K_d measured should be “apparent equilibrium dimer dissociation constant, $K_{d,dimer}$ ” and we have modified the description accordingly.

The reviewer raises an interesting point and it is possible that the value is underestimated slightly as we are titrating unlabelled TLNRD1 against the labelled-TLNRD1. This is why the “apparent equilibrium dimer dissociation constant” description is more appropriate. However, at concentrations lower than, and around, the $K_{d,dimer}$, both the unlabelled and labelled TLNRD1 species will be monomeric (or at least in equilibrium between the two states) and so the underestimation is not likely to be that significant. At concentrations above the $K_{d,dimer}$ you might expect that the dimer of the unlabelled would be an issue, but as seen in the figure, the labelled species is fully dimerised at unlabelled concentrations $\ll 1 \mu\text{M}$ so the dimerisation is tight.

2) “The low speed cosedimentation assay data (Figure 4B/C) are insufficiently annotated: it is not clearly stated that we are looking at pelleted actin; band intensities do not correspond to % below, and there is a different actin concentration from that stated in methods.”

We thank the reviewer for this feedback, and we have improved the description of this section.

In Supplementary Figure S2E we include additional data for the 4H in the bundling assay, showing the low-speed pellet fractions alongside the corresponding supernatant fractions to highlight that the 4H protein is well behaving (as part of the response to Reviewer 3).

Thank you for alerting us to the typo on the actin concentration (it was correct in the figure and method but was incorrect in the figure legend). This error is now rectified.

“Perhaps, more careful quantification/data representation (graphs) might point at a role of 5H in actin binding. The SDS-PAGE data can be then moved to SI.”

We now include improved Electron Microscopy that shows that the 4H bundle also bundles actin alone. We agree with the reviewer that the 5H might tune this interaction slightly but this effect appears to be small as both the 4H and the FL-TLNRD1 bind actin tightly.

3) “The actin cosedimentation bundling assay was done for both FL and 4H constructs, but the visualization (Figure 4D/E) was limited only to the FL protein. Obtaining analogous data for at least negative-stain EM might provide a better insight into the role of the 4H core in actin bundling.”

This was an excellent suggestion, and we have worked hard to improve the EM visualisation. This experiment has only become possible since the labs reopened in March 2021.

We now include improved EM data of the FL, and also the 4H and the F250D (now Figure 4D-F). This provides visual, striking evidence of the interactions of TLNRD1 with actin. We find that the 4H bundles actin similarly to the wild type. The bundles are tight and well ordered.

This set of experiments was also very useful for the F250D monomeric mutation, as this protein clearly binds to the actin filaments showing its actin binding is functional (Figure 4F). However, this protein decorates single actin filaments in stark contrast to the wild type which drives bundle formation.

We also now expand the data on the 4H actin bundling and include the complete low-speed actin cosedimentation bundling assay with the 4H construct (in Supplementary Figure S2E). This data nicely demonstrate the interaction with actin, and the bundling capacity of the 4H domain. It also shows that at higher concentrations of 4H free 4H is detected in the supernatant fractions showing that this interaction is saturatable and that the 4H behaves nicely in solution.

Reviewer #2 (Comments to the Authors):

“The report by Cowell et al., describes biochemistry and structural characterization of TLNRD1, accompanied by basic cell biological assays.

TLNRD1 is a 2 helical-bundle containing protein that has a high similarity to the R7-R8 domain of the focal adhesion protein talin.

The authors found that TLNRD1 dimerizes through the 2nd module helix bundle, particularly through hydrophobic interaction of F250. In addition, they observed actin bundling, which depends on F250-mediated TLNRD1 dimerization.

Cell biological experiments validated the relevance of residue F250 and TLNRD1's filopodia promotion.

Overall, the results of the study are robust and the experiments are carried out thoroughly.”

Thank you for this positive assessment and the useful suggestions.

1. “The side-by-side comparison to the function of talin R7-R8 should also extend to full-length talin. Recently Dedden et al. (2019), reported that R7R8 is inhibited from actin binding in its autoinhibited form. Therefore, it is important to state that R7-R8 may function differently in the full length context compared to a truncated version. The 3D architecture of talin is relevant for the recruitment process and also impacts the function and accessibility of R7-R8. This has an implication for talin R7R8, as it would not act as a dimerization module when incorporated in the structural core of talin full-length, while TLNRD1 can work as an independent isolated module.”

The reviewer raises the issue of a major difference between, the R7R8 domains of talin and the equivalent domains in TLNRD1. The Dedden et al. paper clearly shows that the R7R8 in talin are subject to additional levels of autoinhibition as a result of the additional domains in talin. These domains are not present in TLNRD1 so it can behave as an independent isolated module as the

reviewer highlights. We have included this interesting point in the discussion (page 11) and cited the Dedden et al. paper in the manuscript (page 11).

We have highlighted the binding sites on the FL-TLNRD1 structure (based on Talin complex structures and models) to show that the corresponding residues in TLNRD1 are constitutively accessible and those in talin are tighter regulated. This is now shown in Figure 3H.

2. “I am curious if talin R7R8 would make an interaction with TLNRD1, possibly to form a heterodimer?”

We were also curious of this possibility. However, we were not able to observe any dimerisation, and we did not detect TLNRD1 localisation at focal adhesions which might be expected if the two interacted in this manner.

The comparison of the dimerisation surface of TLNRD1 and the equivalent region of R8 show that they are not compatible. The pocket where the F250 aromatic ring docks in to the other subunit in the dimer, is absent and charged in the talin R8.

We and others have crystallised the talin R7R8 module a number of times (PDBs 2x0c (R7R8 apo), 4w8p (R7R8:RIAM), 5fzt (R7R8:DLC1), 5ic0 (R7R8R9), 6twm (R7R8:CDK1) and we have never detected any evidence of a dimerisation of talin mediated through R8, so we also do not think talin can dimerise in this manner.

3. “The discussion is too lengthy for the contents of this report and should be more concise. The authors give too many speculations for the presented data. Aggressive cancer cell formation seems out of place. The study describes the function of filopodia formation and does not go beyond localization.”

The reviewer raises a good point which is well taken. We have edited the discussion to make it much more concise and to be more representative of the results and to be less speculative.

4. “In the introduction the authors stated TLNRD1 may impact talin R7-R8 binding partners and therefore talin's function as well. Although the biochemical basis of TLNRD1 interacting with R7-R8 binding proteins is shown, cellular experiments are not convincing enough to support this notion. It should be removed from the introduction or extended by further experimental evidence.”

The reviewer makes a valid point, and we have removed the speculation from this statement in the introduction.

Reviewer #3 (Comments to the Authors):

“This manuscript investigates the relatively poorly studied protein TLNRD1, confirming its expected structural organization, dimerization, F-actin binding and interactions with other talin-binding partners but also revealing an unanticipated mode of dimerization and an impact on myosin X stimulated filopodia formation. Data show that TLNRD1 adopted the predicted 5+4 helical bundle architecture but notably also dimerizes via the 4-helical bundle and this appears to support F-actin bundling. TLNRD1 localizes at stress fibers and in cells stimulated to produce filopodia through over-expression of myosin X it localizes at filopodia tips. Filopodia localization and the ability of TLNRD1 to stimulate filopodia formation seems to depend on its dimerization although this conclusion rests on use of a single point mutant that also reduces thermal stability. It is unclear whether TLNRD1 is bundling actin at filopodia tips or is recruited there by other interactions.”

“The authors report crystal structures of both domains of TLNRD1 or of the 4 helical bundle in isolation. The isolated 4-helix bundle structure is not clearly presented in the manuscript but seems to be very similar to the longer structure and to dimerize in a similar manner.”

Figure 1F shows the 4-helix structure overlaid with the full-length structure. We have altered the wording of the figure legend to make this clearer. The 4-helix on its own superimposes onto the 4-helix in the full-length molecule exactly. The structure is identical even down to the buried waters in the interface.

“The dimerization mode is unexpected and is supported by mutagenesis at one residue but this dimerization disrupting mutation also leads to a dramatic reduction in thermal stability - this is acknowledged but care is needed in ascribing the functional consequences of this mutation to a lack of dimerization rather than a lack of stable folding. If other residues along the dimer interface are mutated do they have similar effects on both thermal stability and dimerization, or can they separate the two properties?”

The F250 is the major determinant of the interaction, with the two aromatic rings of the phenylalanine from each monomer docking into the opposing molecule. The inclusion of the reduction in thermal stability was important to provide an honest evaluation of the mutation. The protein is still fully folded at 45°C and we have done extensive biochemical validation of the mutant, both in the original manuscript and in the new data.

Interestingly, the R8 domain of talin has a melting temperature of 37.7°C, and talin R7R8 has a melting temperature of 53°C. And so the 48°C melting temperature of the mutant is more in-line with the thermal stability of most of the talin domains. And it is the dimerisation that is leading to enhanced stability.

However, to further validate the mutant we have performed additional experiments that have been added to the new version of our manuscript:

- NMR analysis of the WT and F250D mutant to show both domains have the same fold and are monomeric in solution (Supplementary Figure S2B).
- FP assay comparing the WT and F250D mutant to show both forms still can bind RIAM, and that the LD binding surface is present in both (Supplementary Figure S2D).
- Perhaps most striking, the new EM data clearly shows that the F250D mutant decorates actin filaments (Figure 4F).

This data complements the data already in the paper

- CD analysis showing that the domain is correctly folded and unfolds cooperatively (Supplementary Figure S2A). The domain is still behaving nicely, the stability at >45°C is just reduced.
- The SEC-MALS data showing a single tight peak that has the exact molecular weight of the monomer species (Figure 2C). There was no protein in the void, so we are confident it behaves well.
- Actin cosedimentation data showing that the F250D still binds actin (Supplementary Figure S2G,H).

This mutant has an effect on thermal stability, but it is our assertion that this is not the mutation itself destabilising the domain (in a monomeric form the phenylalanine is a surface residue with its sidechain completely exposed). The new NMR data we have collected confirms that the domain is still folded as the spectra of the WT and F250D proteins are comparable. Instead, we think the

reduced stability is because the TLNRD1 4helix domain exists as a constitutive dimer (as evidenced by the nM affinity with itself) and is less stable when in isolation, as it always exists as an 8-helix dimeric bundle. Any mutation that disrupted dimerisation would also reduce the stability.

“The authors use FP assays to show that TLNRD1 retains the binding activities of talinR7R8. These data would be enhanced by a direct comparison of the TLNRD1 and talinR7R8 binding results - are affinities comparable?”

Thank you for this useful suggestion. We now include the equivalent talin R7R8 binding experiments in the new version of our manuscript. The FP assays of the talin R7R8 interactions with RIAM and KANK are now added in Figure 3D and 3G and allow direct comparison of the TLNRD1 and talinR7R8 binding results.

“In addition, the authors should comment on the availability of the binding sites in the TLNRD1 structure based on comparison with the talin complexes.”

This is a useful suggestion, thank you. We now include a structural model of TLNRD1 in complex with KANK and RIAM based on comparison with the talin complex crystal structures to illustrate the availability of the binding sites.

“The F-actin co-sedimentation data in Fig 3G lacks controls. This is particularly relevant for the 4H construct which exhibited unanticipated actin binding. Negative controls that do not co-precipitate are needed, as is evidence that the protein is not present in the pellet in the absence of F-actin. Again, a direct parallel comparison with the R7R8 protein might be helpful. Co-sedimentation assays can be used to estimate affinity of the interaction with F-actin and this should be performed, allowing comparison of the wild-type and F250D mutants to ensure that the defective bundling does not relate to reduced affinity for F-actin.”

We have improved the actin binding data and support our data with the direct visualization of the 4H complexation with actin using EM (Figure 4E).

However, we also now include these data:

- the 4H on its own in absence of actin in Figure 3 (now Figure 3I).
- the gel showing the “supernatant” of the bundling assay with the 4H as this is the most striking example of the correct behaviour of the 4H in the assay (Supplementary Figure S2E). At high concentrations of 4H we saturate the binding to actin, which results in free 4H being present in the supernatant fractions. Then at low concentrations of 4H we have almost all actin in the supernatant. This gel clearly demonstrates that the 4H is well behaving in vitro and that the 4H we see in the pellet is due to its interaction with actin.

“The importance of dimerization for bundling is only shown in the TEM image, the quantitative assays (Fig 4B&C) should be repeated to more clearly show the importance of dimerization. As already mentioned, testing additional mutations along the dimerization interface would also be helpful. Identification of mutations that prevent F-actin binding would also be very helpful but may lie beyond the scope of this work.”

We have now expanded the actin bundling data. Our new EM data clearly demonstrates the striking effect the F250D mutation has on TLNRD1 bundling of actin. TLNRD1-FL strongly bundles actin (Figure 4D) compared to the TLNRD1-F250D which binds tightly to the actin filaments but is seen in

the EM to decorate single actin filaments with no evidence of bundling present (Figure 4F). This difference, as a result of the single F250D point mutant rendering TLNRD1 monomeric, provides visual evidence that the bundling is dimerisation dependent. This visual verification of the role of dimerisation in actin bundle formation in the new revised Figure 4 highlights the importance of dimerisation in actin bundling.

This data are strongly supportive of the low-speed actin cosedimentation data already in the manuscript which shows that F250D binds to actin but does not bundle it (Supplementary Figure S2G-H).

We appreciate the suggestion that inclusion of additional mutants, both along the dimerization interface and to explore the actin binding, would potentially provide additional information on TLNRD1 function. However, in light of the very clear data obtained with the F250D mutant, we feel that these would lie beyond the scope of this study.

“The observation that TLNRD1 influences filopodia is interesting but as this is apparently always assessed in MYO10 over-expressing cells its general significance is hard to determine.”

The reviewer raised a valid point, and we have now added additional cell biology data to test the effect of TLNRD1 on endogenous filopodia in two cell lines in the absence of MYO10 over-expression. In both cases, increased expression of TLNRD1 wildtype increases the number of filopodia while this is not the case when TLNRD1 F250D is expressed (new figure 5F-I).

June 10, 2021

RE: JCB Manuscript #202005214R

Dr. Benjamin Thomas Goult
University of Kent
School of Biosciences,
14/21 Ingram Building
Canterbury, Kent CT2 7NJ
United Kingdom

Dear Dr. Goult:

Thank you for submitting your revised manuscript entitled "Talin Rod Domain-Containing 1 (TLNRD1) is a novel actin-bundling protein which promotes filopodia formation". We've now had a chance to assess your revisions and we would be happy to publish your paper in JCB pending final revisions necessary to meet our formatting guidelines (see details below). We hope that you will agree that the rigorous JCB reviewing has resulted in a superior manuscript about which you can be justifiably proud.

A. MANUSCRIPT ORGANIZATION AND FORMATTING:

Full guidelines are available on our Instructions for Authors page, <https://jcb.rupress.org/submission-guidelines#revised>. **Submission of a paper that does not conform to JCB guidelines will delay the acceptance of your manuscript.**

1) Text limits: Character count for Reports is < 20,000, not including spaces. Count includes title page, abstract, introduction, results, discussion, and acknowledgments. Count does not include materials and methods, figure legends, references, tables, or supplemental legends. You are slightly below this limit but please bear it in mind when revising (if you need to go slightly over this limit, it should be fine but please try to be as concise as possible).

2) Figure formatting: Scale bars must be present on all microscopy images, including inset magnifications. Please add scale bars to at least one of the magnified inset images in figure 5A and 5F. Molecular weight or nucleic acid size markers must be included on all gel electrophoresis, including cropped images. Thus, please add weight markers to the blots on figures 4B and C and Supplementary figure 2E, G, and H.

3) Statistical analysis: Error bars on graphic representations of numerical data must be clearly described in the figure legend. The number of independent data points (n) represented in a graph must be indicated in the legend. Statistical methods should be explained in full in the materials and methods. For figures presenting pooled data the statistical measure should be defined in the figure legends. Please also be sure to indicate the statistical tests used in each of your experiments (both in the figure legend itself and in a separate methods section) as well as the parameters of the test

(for example, if you ran a t-test, please indicate if it was one- or two-sided, etc.). In figure 5, you indicate that randomization/permutation tests were used but it is unclear whether this was also used for the analyses in SFig 3C. The statistical tests and parameters must be fully described in a separate methods section. Also, while the randomization test is non-parametric, if you used any parametric tests in your study, please indicate if the data distribution was tested for normality (and if so, how). If not, you must state something to the effect that "Data distribution was assumed to be normal but this was not formally tested."

4) Materials and methods: Should be comprehensive and not simply reference a previous publication for details on how an experiment was performed. Please provide full descriptions (at least in brief) in the text for readers who may not have access to referenced manuscripts. The text should not refer to methods "...as previously described."

5) Please be sure to provide the sequences for all of your primers/oligos and RNAi constructs in the materials and methods. You must also indicate in the methods the source, species, and catalog numbers (where appropriate) for all of your antibodies.

6) Microscope image acquisition: The following information must be provided about the acquisition and processing of images:

- a. Make and model of microscope
- b. Type, magnification, and numerical aperture of the objective lenses
- c. Temperature
- d. imaging medium
- e. Fluorochromes
- f. Camera make and model
- g. Acquisition software
- h. Any software used for image processing subsequent to data acquisition. Please include details and types of operations involved (e.g., type of deconvolution, 3D reconstitutions, surface or volume rendering, gamma adjustments, etc.).

7) References: There is no limit to the number of references cited in a manuscript. References should be cited parenthetically in the text by author and year of publication. Abbreviate the names of journals according to PubMed.

8) Supplemental materials: There are strict limits on the allowable amount of supplemental data. Reports may have up to 3 supplemental figures. At the moment, you meet this limit but please bear it in mind when revising.

Please also note that tables, like figures, should be provided as individual, editable files. A summary of all supplemental material should appear at the end of the Materials and methods section.

9) eTOC summary: A ~40-50 word summary that describes the context and significance of the findings for a general readership should be included on the title page. The statement should be written in the present tense and refer to the work in the third person. It should begin with "First author name(s) et al..." to match our preferred style.

10) Conflict of interest statement: JCB requires inclusion of a statement in the acknowledgements regarding competing financial interests. If no competing financial interests exist, please include the following statement: "The authors declare no competing financial interests." If competing interests are declared, please follow your statement of these competing interests with the following statement: "The authors declare no further competing financial interests."

11) A separate author contribution section is required following the Acknowledgments in all research manuscripts. All authors should be mentioned and designated by their first and middle initials and full surnames. We encourage use of the CRediT nomenclature (<https://casrai.org/credit/>).

12) ORCID IDs: ORCID IDs are unique identifiers allowing researchers to create a record of their various scholarly contributions in a single place. At resubmission of your final files, please consider providing an ORCID ID for as many contributing authors as possible.

B. FINAL FILES:

Thank you for your attention to these final processing requirements. Please revise and format the manuscript and upload materials within 7-14 days. If complications arising from measures taken to prevent the spread of COVID-19 will prevent you from meeting this deadline (e.g. if you cannot retrieve necessary files from your laboratory, etc.), please let us know and we can work with you to determine a suitable revision period.

Please contact the journal office with any questions, cellbio@rockefeller.edu.

Thank you for this interesting contribution, we look forward to publishing your paper in Journal of Cell Biology.

Sincerely,

Kenneth Yamada, MD, PhD
Senior Editor
The Journal of Cell Biology

Tim Spencer, PhD
Executive Editor
Journal of Cell Biology